# Tree Nuts and Peanuts as a Source of Beneficial Compounds and a Threat for Allergic Consumers: Overview on Methods for Their Detection in Complex Food Products

**DOI:** 10.3390/foods11050728

**Published:** 2022-03-01

**Authors:** Anna Luparelli, Ilario Losito, Elisabetta De Angelis, Rosa Pilolli, Francesca Lambertini, Linda Monaci

**Affiliations:** 1Institute of Sciences of Food Production, National Research Council (ISPA-CNR), Via G. Amendola, 122/O, 70126 Bari, Italy; anna.luparelli@uniba.it (A.L.); elisabetta.deangelis@ispa.cnr.it (E.D.A.); rosa.pilolli@ispa.cnr.it (R.P.); 2Department of Chemistry, University of Bari “Aldo Moro”, Via E. Orabona 4, 70126 Bari, Italy; ilario.losito@uniba.it; 3SMART Inter-Department Research Center, University of Bari “Aldo Moro”, Via E. Orabona 4, 70126 Bari, Italy; 4Barilla G. R. F.lli SpA, Analytical Food Science Research, Via Mantova 166, 43122 Parma, Italy; francesca.lambertini@barilla.com

**Keywords:** food allergy, hidden allergens, nuts, peanuts, liquid chromatography-mass spectrometry, markers

## Abstract

Consumption of tree nuts and peanuts has considerably increased over the last decades due to their nutritional composition and the content of beneficial compounds. On the other hand, such widespread consumption worldwide has also generated a growing incidence of allergy in the sensitive population. Allergy to nuts and peanuts represents a global relevant problem, especially due to the risk of the ingestion of hidden allergens as a result of cross-contamination between production lines at industrial level occurring during food manufacturing. The present review provides insights on peanuts, almonds, and four nut allergens—namely hazelnuts, walnuts, cashew, and pistachios—that are likely to cross-contaminate different food commodities. The paper aims at covering both the biochemical aspect linked to the identified allergenic proteins for each allergen category and the different methodological approaches developed for allergens detection and identification. Attention has been also paid to mass spectrometry methods and to current efforts of the scientific community to identify a harmonized approach for allergens quantification through the detection of allergen markers.

## 1. Introduction

### 1.1. Nut/Peanut Allergy: Prevalence and Epidemiology

Food allergy is an adverse immunological reaction that affects genetically predisposed individuals, upon ingestion of specific food constituents; it may cause immediate or delayed reactions, sometimes serious, or even lethal [1].This pathological immune system disorder has become a global issue in recent decades, and it is estimated to affect 5% of adults and at least 8% of children in Western countries, leading to limitations in daily life and creating a heavy burden on the health care system [2]. Allergy to tree nuts and peanuts has a particular relevance in terms of risks for human health, since it accounts for 70–90% of deaths related to food-induced anaphylaxis [3]. Peanut allergy affects 5% of adults and 8% of children in the United States [4], while as for nut allergy, an approximate estimation of 4.9% of sensitive individuals has been reported [5]. Although tree nuts and peanuts are not botanically related, similar clinical symptoms were observed after their ingestion, such as cutaneous, gastrointestinal, and respiratory manifestations [6,7]. Moreover, approximately 30% of people suffering from peanut allergy showed a cross reactivity with tree nuts and about 15–30% of individuals sensitive to a tree nut species was found to show cross reactivity with at least one additional tree nut [8]. In the light of this, cross-reactivity phenomena have been studied to make accurate predictions of possible allergic reactions [8]. The growing incidence of allergy to nuts is due to their huge consumption worldwide, mainly related to their nutritional properties, health benefits, and organoleptic characteristics. The benefits for human health are mainly attributable to the presence of macronutrients (carbohydrates, proteins, and fats), micronutrients (vitamins and minerals), and various phytochemical compounds, such as phenolic acids and flavonoids, with known antioxidant, antimutagenic, and antitumor properties [9].

In recent years, several studies have centered on genetic, epigenetic, and environmental risk factors related to food allergies, bringing more clarity on these issues, and opening interesting perspectives in terms of improvement of prevention and treatment strategies targeted to individuals at risk [10]. However, risk factors that may influence food allergy or sensitization are countless (e.g., sex, vitamin D deficiency, obesity, reduced consumption of omega-3 fatty acids or of antioxidants, geographical origin, and many others), whereby data and statistics on food allergy prevalence are still limited compared to the extension of the problem [10]. Currently, the avoidance of the allergenic food intake by affected individuals represents the best therapeutic strategy, despite progresses in desensitization regimes [11] and in the development of other therapies [12,13].

### 1.2. European Regulatory Framework

Within the Joint FAO/WHO Food Standards Programme, the Codex Alimentarius Commission (CAC) plays a pivotal role in establishing standard approaches to protect consumer health and promote fair practices in food trade. The matters covered are numerous and among these, the critical issues of labelling and food allergens control are carefully addressed. In this regard, a Codex Alimentarius guidance establishing the code of practice of food allergen management for food business operators was recently published. In addition, in response to requests for scientific advice, an ad hoc Joint FAO/WHO Expert Consultation on Risk Assessment of Food Allergens was created in 2019 advising on validation of the Codex priority allergen list through risk assessment, advising on establishing threshold levels in foods for the priority allergens, and reviewing the precautionary allergen labelling (PAL). In 2011, the European Regulation 1169/2011 entered into force, establishing the mandatory indication of 14 allergen classes and derived products in the ingredient list of foods whenever added as ingredients [14]. Peanuts (*Arachis hypogaea*) and a group of tree nuts, specifically almonds (*Prunus dulcis*), hazelnuts (*Corylus avellana*), walnuts (*Juglans regia*), cashews (*Anacardium occidentale*), pistachios (*Pistacia vera*), pecan nuts (*Carya illinoinensis*), Brazil nuts (*Bertholletia excelsa*), and macadamia or Queensland nuts (*Macadamia ternifolia*) were included in such priority list to be indicated on the food label. However, any phenomena producing an unintentional presence of allergenic components to food, as consequence of cross-contamination with other food ingredients (e.g., during manufacturing or packing), are not covered by this regulation. To overcome this limitation, food manufacturers apply voluntary Precautionary Allergen Labelling (PAL) in case of potential unintended presence of allergens to inform and protect allergic consumers [15,16]. However, the lack of a regulatory framework for hidden allergens management and the absence of legal action thresholds have led the food industry to make excessive use of PAL, with consequent reduction of consumer trust [17,18]. Indeed, precautionary phrases have reached such ubiquity that consumers have often been reported to ignore them [19]. On the other hand, defining guidelines to manage the presence of hidden allergens in food products on a quantitative basis is a very complicated task. The broad inter-individual variability range in terms of minimum amounts of allergens eliciting an adverse reaction, from a few milligrams to a few grams, and of sensitivity to different epitopes of the same protein [20] contribute to the difficulty in setting up a regulatory framework.

Recently, various European countries such as Switzerland, Germany, Belgium, and the Netherlands have defined legal thresholds, but considerable disparity is observed among them. In Australia and New Zealand, the Voluntary Incidental Trace Allergen Labeling (VITAL) system defines eliciting doses (EDs), which, however, do not have a regulatory status, based on clinical studies for the protection of 95% of allergic people (ED05) or 99% of the population (ED01) [17,21,22]. After the first set of thresholds published in 2011, systematic data collection has continued from 2011 to 2018, resulting in an updated dataset of clinical data for fourteen allergens [23]. According to the new data, the Panel recommends the adoption of ED01 values as the Reference Doses for VITAL 3.0 and set at 0.2 mg of total protein for peanut, 0.03 mg of total protein for walnut, 0.1 mg of total protein for hazelnut, and 0.05 for cashew (Allergen Bureau, Summary of the 2019 VITAL Scientific Expert Panel Recommendations). Although these reference doses have not been standardized yet, these are useful as indicative thresholds for laboratories and food authorities to support some decisions, such as for food recalls. As can be easily inferred, the development of increasingly sensitive, rapid, and reliable analytical techniques, able to identify and quantify allergenic components in food products even at low concentrations, are demanding in order to support the risk assessment for allergen management.

In this review, an overview of analytical methods for monitoring food allergens, with a specific focus on peanuts and five nut species (almonds, hazelnuts, walnuts, cashews, and pistachios), is presented.

### 1.3. Analytical Methods for Protein Allergen Detection in Foods

From an analytical point of view, several methods including immunochemical tests, molecular biology techniques, and approaches based on mass spectrometry coupled to chromatography have been developed for tracing allergens in food. Most of them rely on the detection of allergen proteins contained in the offending food. Specifically, ELISA (Enzyme-Linked Immuno Sorbent Assay) and PCR (Polymerase Chain Reaction) are, respectively, the immunochemical and molecular biology techniques mainly used for allergen detection. ELISA represents a widely exploited technique for allergen detection as it is reasonably quick and targets the allergenic protein itself through the binding with a specific receptor (i.e., an antibody) [24]. The ELISA analytical workflow is relatively rapid, inexpensive, and simple enough not to require trained laboratory personnel. To date, several ELISA-based allergen detection kits are commercially available for the main food allergens. On the other hand, numerous critical issues could affect the final response of ELISA tests, such as cross-reactivity phenomena that can produce false positives or hook effects, which are responsible for false negatives results [25,26]. In the modern society, foods are subjected to several different technological processes before their consumption, and if, on one hand, these practices improved the organoleptic or nutritional value of some foods, on the other they could affect the native protein structure/conformation promoting their denaturation and/or modifications [27]. Food processing could also influence protein extractability due to the formation of protein aggregates [28]. Such phenomena affecting the protein structure could result in a loss of the conformational epitopes, thus destroying the ELISA recognition sites with consequent production of false negative results [24]. On the contrary, false positive results could be obtained upon cross reactivity between specific antibodies used in the ELISA test with unknown substances of the food matrix [29].

Among molecular biology methods, PCR technique based on the amplification of total genomic DNA in a food product has been extensively exploited in the analysis of food allergens. Although conventional PCR has been used mainly for qualitative determination, the latest advances in real time-PCR have enabled the semi-quantitative/quantitative determination of protein allergens in food matrices, with interesting applications in routine analysis [30,31,32,33,34,35,36,37,38]. As for DNA-based detection techniques, specific functional molecular markers are obtained not for the “direct” protein allergen identification, but for the identification of the gene coding for it, and thus it is less informative for sensitive individuals [39]. In fact, allergenic DNA in food can be inoffensive to allergy sufferers if no allergenic proteins are present, but at the same time, allergenic proteins may be present even if allergenic DNA is absent. DNA-based methods are a viable alternative for foods for which ELISA tests are not yet available or for food matrices whose allergens are not detectable by ELISA. However, there are limitations that influence this approach: false results could be produced in the case of food submitted to technological process (poor DNA recovery from samples) or in the case of presence of interferences and enzyme inhibitors (giving rise to false negatives) [15].

Due to its potential ability to overcome the aforementioned limits, mass spectrometry (MS) has become increasingly relevant in allergen analysis. MS relies on the identification of a target molecule by its molecular mass, offering, at the same time, a good detection sensitivity, essential to uniquely identify proteins present as traces in complex matrices. Most of analytical methods based on MS platforms for food allergens detection use the *bottom-up* approach for allergen proteins identification (Figure 1).

More specifically, in the *bottom-up* approach, the food protein extract is digested by means of suitable proteolytic enzymes, generating a mixture of peptides that contain the signature-peptides of the target molecules, namely the allergenic proteins. Peptides resulting from the enzymatic cleavage of the proteins (trypsin is the most widely used endopeptidase) can be analyzed either after in solution digestion of the whole extract, i.e., by MALDI coupled to high resolution MS, or can be preliminarily separated by liquid chromatography (LC) and, after soft ionization (ESI), detected by MS or MS/MS analysis.

Specifically, two different strategies could be implemented for protein/peptide characterization and identification, namely peptide mass fingerprinting (PMF) and peptide fragmentation fingerprinting (PFF). As for PMF, the identification of proteins was obtained by comparing the experimental peptide masses with the theoretical ones present in the sequence database; for PFF, the database search focuses on the precursor and fragments masses produced by MS/MS events of specific peptides resulting in a more reliable identification [40,41,42].

Bioinformatics software and online databases are often used for the characterization and identification of peptides obtained by comparing experimental MS data and calculated mass values returned from sequence databases processed by the search engine (e.g., Mascot, MS-Fit, ProFound, MassSearch, Sequest etc.) [43]. The quantification of proteolytic peptide(s), used as a surrogate marker(s) for the precursor protein, shows several advantages such as specificity, sensitivity, and a wide dynamic range that can cover four or five orders of magnitude [15].

### 1.4. Untargeted Versus Targeted Mass Spectrometry for Allergens Analysis

*Bottom-up* proteomics can be considered the most widely used approach for characterization and identification of allergenic proteins and for allergenic traces quantification in complex food matrices [30,44]. For developing an analytical method aimed at allergens quantification, a non-targeted MS/MS analysis is initially performed without any pre-selection or prefiltration of precursor ions, the only filter being the intensity of the ion which should exceed a threshold to be fragmented. This preliminary step helps in the identification of peptides detected by MS where the most reproducible and intense peptides can be identified and proposed as reliable allergen markers. Peptides detected by MS are typically identified by means of bioinformatics tools and only those fulfilling specific criteria described in the review by Pilolli et al., 2020, are shortlisted as candidate peptide markers for the associated allergenic protein [45]. The selected peptides are fragmented, and specific transitions are monitored to increase method sensitivity. Peptide-based allergen quantification strategy generally relies on Selected Reaction Monitoring (SRM) acquisition mode or by Multiple Reaction Monitoring (MRM) acquisition mode, where multiple transitions are monitored simultaneously.

Selected ion monitoring scheme (SIM) acquisition mode was used for the analysis of food allergens as well by exploiting ESI-Q-TOF MS systems coupled with UHPLC or micro or nano HPLC separation [30,42,46,47] confirming this approach as an alternative confirmatory method for the detection of food allergens despite the limited sensitivity offered compared to the SRM mode. It is worthy to be noted that the robustness and sensitivity of a targeted MS-based method strictly depend on the selection of the most appropriate proteotypic peptides that highlight the presence of the allergenic protein to which they are associated. Recently an overview of the current literature on MS-based analytical methods for allergen detection was performed and several critical aspects affecting the reliability of peptide markers were discussed [45]. In the light of such considerations, the criteria for the selection of the most reliable peptides for allergen proteins detection were reported [45].

Through the introduction of the most recent highly sensitive triple quadrupole mass spectrometers, tandem mass spectrometry has been proposed for the development of multi-allergen methods based on the MRM MS operating mode [48,49,50]. Absolute quantification of protein allergens based on the MS detection of their peptides, however, requires adequate calibration and standards (isotopically labelled form of the target analyte) to be used as internal standards to compensate for the ionization reproducibility. The influence of matrix components on tryptic digestion efficiency and peptide LC-MS signal abundance or peptide stability have been reported elsewhere [51,52,53,54].

In proteomics, commonly used quantitative applications of SRM are based on the principles of stable isotope dilution (SID) methods, often suggested for absolute quantification. A very interesting protein quantification, with LOD in sub-ppm range [55], can be achieved through the use of stable isotope-labeled standards such as AQUA peptides, QconCAT (concatenated peptide constructs), and PSAQ (protein standard absolute quantification). A different quantitative technique is known as label-free quantification, in which MS1 ion current or MS2 spectral count is used to identify differentially abundant peptides. The most used procedures for quantitative analysis of allergens, in this case, are external calibration [56,57], the use of synthetic peptides [58], or standard addition [59].

As alternative, a non-targeted HR-MS approach based on Orbitrap technology has been proposed for a rapid and high-throughput screening of allergens in different food matrices [50,60]. This technique allows to identify a food component and to know its chemical structure and, as a non-targeted approach, simultaneously identify numerous peptide markers, even retrospectively without the request for preliminary information/selection [61]. Furthermore, the combination of HR-MS full scan and All Ion Fragmentation acquisition mode (FullMS/AIF) at maximum resolving power and mass accuracy can provide confirmatory and quantitative characteristics within a single chromatographic run [62].

Thanks to the many advantages offered with respect to the immunological and molecular methods for allergen analysis, the MS-based approach is eligible to become the reference method for food allergen analysis. In particular, among MS methods, the SRM show the multiplexing capacity required for a multiallergen screening tool and the potential to provide a rigorous method for allergen analysis [15]. In light of this, many efforts are directed to develop MS-based reference methods for detecting allergens in accordance with the performance criteria of the Standard Method Performance Requirements (SMPR, 2016) established in 2016 by AOAC International for allergens detection as well as the performance criteria on analytical methods and interpretation of data established by Council Directive 96/23/EC about the monitoring of substances and residues thereof in live animals and animal products. Anyway, different tricky issues hinder the pursuing of this goal, above all the question of the “common measurand” of the methods, that is neither the “content of the individual proteins” nor the “total food”, but the “sum of the total proteins from a particular allergenic food ingredient in a whole food” [63]. Therefore, analytical methods are needed that provide results to be presented in milligrams of total allergenic protein per kilogram of food rather than on a commodity basis are required. In fact, to carry out the risk assessment on which the reference doses of allergenic foods are based, affected patients were administered known amounts of the offending total food allergen protein [23,64]. Different analytical MS-based methods for allergen detection have been proposed to date, but the results provided are not always comparable in terms of unit of measure. Therefore, specific conversion of measurement results into the common measurand (reported quantity, mg of protein per kilograms of food) would be needed, as this is the information on which risk assessments and the establishment of clinical thresholds are based on [63]. Another aspect to underline is the implementation of calibration systems in order to guarantee the comparability between analytical results. Since the analyte for allergen analysis are proteins that are not directly measurable, the implementation of a reference measurement method would provide analytical results traceable to the same reference.

In the light of these considerations and given the need for designing a reliable and sensitive method for food allergen analysis, very recently a dedicated project was founded by EFSA (EFSA call GP/EFSA/AFSCO/2017/03) titled “Detection and quantification of allergens in foods and minimum eliciting doses in food allergic individuals”—ThRAll [15]. The project focuses on the development of an MRM reference method for the simultaneous analysis of six allergens food (cow’s milk, soybean, tree nuts, hen’s egg, and peanut) that cause severe IgE-mediated reactions [65,66].

Meanwhile, additional efforts to develop reliable methods for allergens detection have been carried out worldwide. For example, in 2020 a method met the minimum performance requirements of SMPR 2016.002 and approved as First Action by AOAC Official Methods Board for whole egg, whole milk, peanut, and hazelnut in different food matrices was developed and evaluated in a single-laboratory validation by New and co-workers [67]. In the same year, Martinez-Esteso et al. proposed a quantitative reference method for the analysis of milk allergens in sustained baked food at clinically relevant concentration. Among the key factors taken into account for method development were the optimization of proteins extraction, the complete digestion of the extracted proteins and the release of peptides equimolarly, the use of conversion factors to express the number of measured proteins into total milk protein and the estimation of the uncertainty of the final result [63]. Starting from these premises, this review aims to describe the most recent applications of LC-MS methods for the detection and quantification of protein allergens of peanuts and some of the most important nuts, specifically almonds, hazelnuts, walnuts, cashews, and pistachios, in different types of food. The results of studies published for each of the mentioned species will be described in separate sections, after a brief report of information concerning each species in terms of nutrition and health, major protein allergens described so far, and results obtained with different analytical approaches (immunochemistry or molecular biology).

## 2. Overview of Nuts and Peanut Allergens Composition and Analytical Methods for Their Detection

### 2.1. Almond

#### 2.1.1. General Information and Main Allergens

Almond, scientifically known as *Prunus dulcis*, belongs to the *Rosaceae* family and to the *Prunoideae* subfamily. Almonds have a high nutritional value and are rich in healthy components like mono/poly-unsaturated fatty acids, fiber, vitamins, minerals, phytosterols, and polyphenols [68]. In Figure 2 the nutritional values of almond are detailed. Consumption of almonds can lead to cardioprotective effects [69] and glycemic control [70,71]; moreover, their inclusion in low-calorie diets seems to promote weight loss [72] without neglecting the sense of satiety, due to proteins and fibers present in abundance [73]. Polyphenols and skin fibers also have a prebiotic effect on the human gut microbiota [74]. Cultivated almond varieties show a different chemical profile due to genetic and ecological factors and processing conditions.

Unfortunately, almonds also represent a significant concern from an allergological point of view. Indeed, almond allergy is the third most reported nut allergy in the US (9–15% incidence), prevalence in UK ranges from 22% to 33% [5], whereas in South Asia the incidence of children with sensitization and allergy to almonds is 61.9% and 7.4%, respectively [75]. Currently, the World Health Organization and the International Union of Immunological Societies (Subcommittee on Allergen Nomenclature) has registered the following almond allergens: Pru du 3 (a non-specific lipid transfer protein 1-nsLTP1), Pru du 4 (a profilin), Pru du 5 (a 60S acidic ribosomal protein), Pru du 6 (amandin), and Pru du 8 (an antimicrobial protein). Two other almond proteins were classified as allergens due to their sequence homology with other known food allergens and their belonging to protein families with allergenic properties in several species: Pru du 1 (a pathogenesis related-10, PR10, protein), and thaumatin-like protein Pru du 2 [76]. Almond allergy is often associated with allergies to other fruits that may contain labile (PR 10) or stable (LTP) allergens, leading to multiple sensitizations to pollen, fruits, nuts, and other vegetables [77]. Pru du 3 is a nsLTP1 that can cause allergic reactions even without prior sensitization [77] and was added to the WHO-IUIS allergen database in 2009. In 2015 Buhler et al., isolated, sequenced, and identified Pru du 3 from almond and its potential allergenicity was confirmed through an in silico approach [22].

Pru du 4, included in the WHO-IUIS allergen database in 2006, is a profilin present in all eukaryotic cells, hence it is considered a panallergen [76]. Since the level of Pru du 4 in the almond is low, recent studies have focused on optimizing its expression, purification and characterization using recombinant technology [78]. Despite the high risk of sensitization and allergic reactions due to its ubiquity, clinical manifestations following its ingestion or inhalation are mild or completely absent, due to its low resistance to heat and gastric juices [79]. Pru du 5 was added to the WHO-IUIS allergen database in 2007 and is a component of the large 60S subunit of the 80S eukaryotic ribosomes. Further studies are needed on the symptoms related to its ingestion. Pru du 6 is the most studied almond allergen and was added to the WHO-IUIS allergen database in 2010. This allergen protein, also named Amandin or Almond Core Protein (AMP), accounts for 65% of the total protein content and belongs to the cupin superfamily [80]. Pru du 8 is the first member of a disulfide (S–S) rich antimicrobial protein (ssAMP) family that has been identified as a food allergen. It reacted with IgE in 6 out of 18 sera of patients with almond allergies and was added in the WHO/IUIS database as a food allergen in 2018 [81].

It is worth to be noted that since 2019 the draft genome of almond was available thanks to the sequencing work of Sánchez-Pérez and co-workers [82]. Specifically, they sequenced the sweet homozygous almond cv. Lauranne (genome 2*n* = 2*x* = 16, haploid genome size = 246 Mb) with a combination of Illumina (paired-end and 5-kb mate pairs) and PacBio technologies with the final goal to show the genetic differences between toxic bitter almonds and sweet almonds. They found that a mutation in the bHLH2 basic helix-loop-helix transcription factor prevents the amygdalin production, resulting in the sweet almond genotype, which was actively selected for domestication [82]. In 2020, the whole genome of almond cv. Texas was sequenced and compared to other genomes, among which that of *Prunus persica*, identifying in the movement of transposable elements (TEs) the main cause of the phenotipic differentiation between both species [83]. Starting from the whole genome of *Prunus dulcis* cv. Texas, the whole proteome was obtained and now available on Uniprot online database (https://www.uniprot.org/proteomes/UP000327085, accessed on 10 July 2021).

#### 2.1.2. Immunochemical/Molecular Biology Analytical Methods

Several analytical methods are currently used for the detection of potential almond allergens in processed food. Among them, the most widely used is the ELISA test which has the advantages of fast performance (30–35 min), versatility, reliable results, and limit of detection (LOD) up to 0.1 mg/kg of almond protein in food samples [84,85]. Furthermore, on the base of the food matrix investigated and the sandwich ELISA kit tested, LOD in the range of 3–39 mg/kg were obtained by Garber and colleagues [86]. Liu, Changqi, et al. 2017 developed mAb 4C10-based ELISA for almond detection. This in-house assay and commercially available sandwich ELISA kit were similar in sensitivity (LOD < 1 ppm full fat almond, LOQ < 5 ppm full fat almond) and specificity (no cross-reactivity with 156 tested foods at a concentration of 100,000 ppm whole sample) [87]. The method developed by Slotwinski, E. et al. in 2018 allowed the analysis of almond traces by a direct competitive ELISA in vanilla ice cream, biscuits, pasta, and milk chocolate with an LOD of 0.3 mg/kg and an LOQ of 2.5 mg/kg [88]. Although widely used, ELISA kits have been demonstrated to be often unsuitable to accurately determine almond proteins in processed foods due to their proved thermolability [89]. In 2021, Civera, A. et al. developed sandwich ELISA to detect traces of almond in orange juice, coffee liqueur, chocolate, soy drink, salad dressing, rice, ice cream, and goat cheese with an LOD of 0.02 mg/kg and LOQ of 0.12 mg/kg with low cross-reactivity to Chestnut [90]. An overview of the most recent ELISA kit so far developed for almond allergen detection is reported in Table 1.

Among DNA-based methods, real-time PCR has so far been the most widely applied PCR strategy for detecting food allergens like almond [34,77], allowing the detection of low amounts of almond DNA (5 pg), with LOD values between 1 and 100 mg/kg of almonds in biscuits [36]. With the introduction of the nested approach [33] it was possible to reduce the LOD to 50 mg/kg and recent advances, such as droplet digital PCR (ddPCR) and High-Resolution Melt (HRM), represent a step forward in terms of accuracy and speed of analysis. As an example, the application of the HRM allowed to distinguish almond from other fruits of the Rosaceae family [32].

#### 2.1.3. Mass Spectrometry-Based Analytical Methods for Detecting Almond Allergens

Several studies based on the MS detection of peptides originated from almond protein allergens in different food products have been reported in the last decade, with several types of MS instrumentation involved. In 2010 Bignardi et al. [91] effectively implemented a method based on liquid chromatography (LC) coupled to an Electro Spray Ionization (ESI)-MS equipped of linear ion trap (LIT) mass analyzer for the simultaneous detection of cashew (Ana o 2), hazelnut (Cor a 9), almond (Pru du 1), peanut (Ara h 3/4), and walnut (Jug r 4) in cereals and biscuits. The exploitation of a LIT-mass analyzer enabled to identify at high specificity the most appropriate marker peptides for allergen detection thanks to the monitoring of the peculiar product ions resulting from their fragmentation inside the ion trap (Selective/Multiple Reaction Monitoring, SRM/MRM, mode). Additional experiments by implementing sequential mass spectrometry (MS3) acquisitions were accomplished for evaluating the performance of the LC-ESI-LIT-MS method. By focusing on almond, LOD and LOQ of 17 mg/kg and 58 mg/kg, respectively, were obtained for the Pru du 1 protein in breakfast biscuits. Meanwhile, LOD and LOQ values of 25 mg/kg and 80 mg/kg, respectively, were calculated for Pru du 1 in the same matrix by working with MS3 acquisition mode [91].

In 2011 Heick, et al. [48] described an LC-MS/MS method based on triple-quadrupole mass analyzer and MRM acquisition mode for the detection of seven allergens (milk, egg, soy, peanut, hazelnut, walnut, and almond). The use of marker peptides enabled the simultaneous identification of several allergens, namely hazelnut, peanut, walnut, almond, milk, egg, and soy in concentrations ranging from 10 to 1000 mg/kg in incurred bread. In particular, four different marker peptides (see Table 2) were selected for targeting prunin as it is the main almond allergen, and an LOD of 3 mg/kg was obtained considering one of them [48]. Further investigations about almond tracing in different foods were accomplished by Planque and co-workers in 2017. Specifically, a rapid, highly specific, and sensitive LC-MS/MS based on a triple quadrupole MS platform was developed to detect ten allergens, including almonds, in four different matrices (ice cream, sauce, cookie, and chocolate). Challenging LOQ were achieved for this allergen by working in MRM acquisition mode, namely 5 or 25 mg/kg depending on the marker peptide [52].

In recent years, high-resolution mass spectrometry (HRMS), usually based on the Orbitrap mass analyzer, was satisfactorily applied for the detection of allergens in foods, including those related to almond. For example, in 2018 a method for screening multiple allergens in chocolate based on LC coupled to a hybrid quadrupole-Orbitrap mass spectrometer was developed by Gu et al. [92] and LOD of 1–3 mg/kg were obtained for nuts, including almond. The method described has been applied effectively to the analysis of multiple allergens in chocolates of different brands, with the aim of ascertaining any discrepancies between the allergen content and the food allergen labelling [92].

The main allergenic almond proteins and the corresponding marker peptides reported so far in the literature have been reported in Table 2; Table 3 shows an overview of the most recent LC-MS methods developed for the quantification of nuts and peanuts in food.

### 2.2. Hazelnut

#### 2.2.1. General Information and Main Allergens

*Corylus avellana* L., the European hazelnut, belonging to family of *Betulaceae,* is the second most widespread nut in the world after almonds and its production involves North Africa, Europe, Asia Minor, and Caucasus region. Hazelnut plays an important role in terms of nutrition and human health due to their content in proteins, fats (mainly oleic and linoleic acids), dietary fiber, vitamins (especially vitamin E), minerals, phytosterols (mainly β-sitosterol), and phenolic compounds with antioxidants properties (detailed in Figure 3). Hazelnuts typically consists of 62% fat, 16% protein, and 11% carbohydrate, and its composition can change depending on variety [93,94].

In the group of tree nuts, hazelnut has often been reported as causing allergic reactions [95] with symptoms ranging from mild oral to severe systemic reactions [96]. Children with hazelnut allergy often react with generalized symptoms, while adults more frequently experience localized oral symptoms [97,98]. Hazelnut allergy presents a fairly high overall incidence, namely almost 7.2% of the test population [99] and it can be considered the most prevalent tree nut allergy in Europe [100]. In terms of geographical distribution of the prevalence of hazelnut allergy, USA shows the highest percentage with a prevalence of 14.9%, followed by Germany with 14.7%, Norway with 12.8%, Switzerland with a prevalence of 12.6%), and Sweden with 11.8% [99,101].

To date, twelve groups (Cor a 1, Cor a 2, Cor a 6, Cor a 8, Cor a 9, Cor a 10, Cor a 11, Cor a 12, Cor a 13, Cor a 14, Cor a 15, and Cor a TLP) of hazelnut allergenic proteins have been characterized and included in the official allergen list of the International Union of Immunological Societies/World Health Organization (IUIS/WHO) with the exception of Cor a TLP [101].

Specifically, Cor a 8 (9 kDa) and Cor a 14 (14–16 kDa) belong to the prolamin superfamily and correspond respectively to nsLTP and to 2S albumins. These two small proteins can induce moderate to severe/systemic clinical symptoms, being classified respectively as major and minor allergens in hazelnut and have a high stability to heat treatments thus preserving their allergenicity even after processing [102]. Cor a 9 (40 kDa/per subunit) and Cor a 11 (48 kDa/per subunit) belong to the cupin superfamily and correspond to the 11S legumin-type globulins and 7S vicilin-type globulins, respectively. Similarly to Cor a 8, they cause severe and systemic allergic reactions and are among the major allergens. In addition, Cor a 9 and Cor a 11 also maintain their allergenic potential after food processing as they are thermostable [80]. Cor a 1 (belonging to PR-proteins) and Cor a 2 (belonging to profilins) are considered as food allergens and aeroallergens, as they are present in the reproductive tissues such as pollen, fruit, and seeds. Contrary to what observed for allergens above mentioned, these proteins exhibit a thermolabile behavior, thus suggesting that heat treatments occurring during food processing could affect the stability of the molecules also promoting a reduction of their final allergenicity [103]. Cor a 10 belongs to 70 kDa heat shock proteins (Hsp70), a family of molecular chaperones ubiquitously expressed in nature and involved in the expression and structural integrity of several proteins. Finally, Cor a 12 and Cor a 13 have been classified as oleosins structural proteins found in vascular plant oil bodies and in plant cells. Cor a 12 showed a high degree of similarity to allergenic oleosins of other species, such as those found in sesame (50% identity) and peanut (48% identity). Cor a 13 also showed a high degree of homology with almond (73% identity) and maize (55% identity) oleosins but so far, neither has been associated with allergic reactions [103].

#### 2.2.2. Immunochemical/Molecular Biology Analytical Methods for Hazelnut Detection

In recent years, several proteins and/or DNA-based methods have been developed for the specific analysis of hazelnuts in food [103]. Among them, ELISA tests used so far have shown high performance, with observed LOD between 0.03 and 2.5 mg/kg of hazelnut protein in various foods (Table 1). Since the strong influence of the matrix effect on the reliability of ELISA results, different studies were accomplished to evaluate the performance of commercial kits of different brands for detecting traces of hazelnuts in processed foods [86,104]. These studies demonstrated that the tested ELISA kits provided poor performance when used for hazelnut analysis in complex food matrices. On the contrary, in 2014 and 2015 a method using both poly- and monoclonal antibodies was developed which was found to detect hazelnut proteins in chocolate matrix at a 1 mg/kg concentration level [37,105]. The most recent studies investigating ELISA tests for hazelnut analysis are reported in Table 1.

Among molecular approach, Ehlert et al. in 2009 developed a multitarget method based on a ligation-dependent probe amplification (LPA) for the simultaneous identification of 10 allergenic foods achieving, for hazelnut, an LOD of 5 mg/kg and 100 mg/kg in chocolates biscuits, respectively, thus highlighting the suitability of the method also for the analysis of processed foods [106]. Since 2009, different advanced multitarget methods based on real-time PCR have been developed for detecting hazelnut [35,36,38].

In 2021, to evaluate the cross-contamination phenomena and determine the exposure of allergic individuals to products contaminated by hazelnut, 863 samples were analyzed using ELISA tests (RIDASCREEN^®^FAST Hazelnut, r-Biopharm, Darmstadt, Germany). Most hazelnut samples had a PAL for tree nuts/hazelnuts (94%; *n* = 807) with 6% claimed “nut-free” (*n* = 56). Hazelnuts were found in 9% (0.4–2167 ppm) of samples. All results above the LOQ of the kit of each allergen (0.375 ppm for hazelnuts) are considered positive. All results below, even above the LOD were considered without detectable allergens, or negative results (0.03 ppm) [107].

#### 2.2.3. Mass Spectrometry-Based Analytical Methods for Hazelnut Analysis

In the last decade several MS-based proteomics methodologies for the detection of hazelnut allergen in food matrices were successfully developed. In 2010, Arlorio and co-workers investigated the addition of hazelnut oil in extra-virgin olive oils by implementing a matrix-assisted laser desorption ionization coupled to time-of-flight mass analyzer (MALDI-TOF MS) platform. By tracing two oleosin isoforms and Cor a 9, it was possible to detect the addition of 1% of hazelnut oil to extra-virgin olive oils. This adulteration could represent a serious risk for an allergic patient [108]. In the same year, an LC-ESI-LIT-MS/MS-MS3 instrument was successfully employed by Bignardi et al. (2010) for the detection of Cor a 9 allergen in spiked biscuits. LOD of 30 µg/g and 35 µg/g by working in SRM and SRM3 acquisition mode, respectively, were achieved, while LOQ values of 90 mg/kg (SRM) or 110 mg/kg (SRM3) were reported [91].

Heick et al. achieved interesting results in 2011 [48,49] by detecting four Cor a 9 peptides in spiked bread and flour. The LOD achieved was 5 mg/kg in both matrices investigated demonstrating the remarkable sensitivity of the proposed method both for raw and for processed food samples. A comparable LC-MS/MS approach was shown by Ansari et al. using eight different peptides from Cor a 8, Cor a 9, and Cor a 11 proteins [109]. Although sensitivity levels were not reported, the potential application of selected peptides for the identification of hazelnut in processed foods was evaluated. Interestingly, one of the four peptides referred to Cor a 9 was detected also in tryptic digests of proteins occurring in pecan, walnut, pistachio, and cashew, and one of the three peptides selected for Cor a 11 was also related to walnut and pecan, thus emphasizing that the use of single peptides for the univocal identification of hazelnut is to be avoided when the presence of other nuts cannot be excluded [109]. In a recent study by Van Vlierberghe et al. four different processing conditions (heating, low pH, and high fat content environments, induction of the Maillard reaction) were chosen to produce food matrices by simulating the production chain commonly used in the modern food industry [110]. Tryptic digests of allergenic proteins were analyzed by HPLC-ESI-Q-ToF-MS and eight peptides arising from two main hazelnut allergenic proteins, vicilin Cor a 11 and legumin Cor a 9 (see Table 2), were selected for identification and quantification [110].

Throughout the years, more sensitive multi-target MS-based methods for detecting hazelnut proteins along with other common food allergens extracted from processed or unprocessed food matrices has been developed [20,40,50,92,111,112]. LOD and LOQ down to 7 and 20 mg/kg, respectively, were obtained by Pilolli et al. [113] by analysing hazelnut in cookie digests by exploiting a multidimensional chromatographic set-up coupled with LIT based mass spectrometer. In 2018, the same authors investigated the feasibility of a micro-HPLC-HR-MS system provided by a hybrid quadrupole-Orbitrap™-mass analyzer in combination with a sharp protocol for sample preparation to detect hazelnut in cookies [114]. They found that by working in t-SIM/dd2 mode LOD down to 20 mg/kg were obtained for the most sensitive hazelnut peptide, corresponding to 6 ng of matrix injected [114]. Successively, more sensitive LOD and LOQ were obtained by Gu et al. in 2018 [92], namely 0.5 µg/g and 1.7 µg/g respectively, by working with triple quadrupole mass spectrometer and SRM acquisition mode. Very recently, a method for the detection of hazelnut in cookies, in compliance with the Standard Method Performance Requirements (SMPR) for the detection and quantification of selected allergens in 10 food matrices, was proposed by New et al. [67]. Parameters such as method quantitation limit (MQL), method detection limit (MDL), extraction recovery, accuracy, and repeatability precision (RSDr) were evaluated and found to fit with the SMPR guidelines. In particular, a MQL and MDL of 5.3 and 3.1 mg/kg were respectively estimated for detection of hazelnut in cookies while the calculated recovery was comprised between 70 and 77% depending on the calibration standard level analyzed. Currently, this method was approved by the AOAC Official Method Boards as First Action.

Table 3 shows an overview of the most recent LC-MS methods developed for the quantification of nuts and peanuts in food.

### 2.3. Cashew

#### 2.3.1. General Information and Main Allergens

Cashew (*Anacardium occidentale* L.) belongs to the *Anacardiaceae* family, native to Brazil and spread spontaneously in South American countries [115]; it is currently the third most consumed dried fruit in the United States [116]. This is a very nutritious food due to its content in fats, vitamins, fibers, amino acids, sterols, and minerals [117]. Moreover, the thin skin of their kernels is rich in phenolic compounds with antioxidant properties, such as catechin/epicatechin and anacardic acids, with the latter corresponding to hydroxy-benzoic acids with different alkyl substituents on the phenyl ring [118,119]. In addition to the compounds with high antioxidant power, cashews also contain bioactive compounds like oleic acid, linoleic acid, stearic acid, phytosterols, beta carotene, lutein, thiamine, zeaxanthin, arginine, and tocopherols, exerting several beneficial effects for human health [120]. A detailed scheme of the nutritional components of cashew is reported in Figure 4. The consumption of cashews was related to a reduced risk of cardiovascular disease due to the content of monounsaturated and polyunsaturated fatty acids [116]. Not surprisingly, cashews have a long history of application in traditional medicine for the treatment as asthma, diabetes, skin infections and inflammation [121]. Nevertheless, there are no clinical studies confirming its effective medicinal efficacy [122].

On the other hand, the rapid inclusion of cashews in dietary habits has led to an extended sensitization and allergy to this food all over the world [123,124,125,126]. Published reports reveal that cashew allergy is found to a greater extent in children [123,127] and in adult women [128].

Currently, three allergenic groups of proteins have been identified for this species and included in the official list of allergens of WHO/IUIS. Among them, two belong to the cupin (Ana o 1 and Ana o 2) and one to the prolamin (Ana o 3) superfamily of proteins. Specifically, Ana o 1 is a vicilin, Ana o 2 an 11S globulin similar to that found in legumes, and Ana o 3 is a 2S albumin.

Ana o 1 is resistant to heat and proteolysis [129] and, despite what was observed by De Leon, M. et al. in their previous study [130], presented cross-reactivity between Ara h 1 and Ana o 1 allergens through analysis with human sera of peanut-allergic patients [131]. A severe cross-reactivity of cashew was also found with pistachio (also belonging to the *Anacardiaceae* family) in patients allergic to cashew and/or to pistachio [128]. This clinical evidence was confirmed by a high overall identity (80%) and similarity (90%) between the sequences of proteins Ana o 1 and Pis v 3 (pistachio), the latter sharing high homology in two of the Ana o 1 immunodominant peptides [132,133]. Cashew appears to be the primary sensitizing agent in some cashew/pistachio allergenic patients [133].

Ana o 2 (also present in the Ana o 2.0101 isoform) is an allergenic legumin 457 aa long and with a molecular weight approximately 55 kDa. This protein shares the sequence identity and similarity for a percentage between 42 and 74%, with several allergenic 11S globulins, that is with Cor a 9 of hazelnut, Jug r 4 of walnut, Ara h 3 of peanut, Ses i 7 of sesame, and Gly m 6 of soybean [134,135].

In 2002, Teuber et al. isolated three small proteins with molecular size below 14 kDa, classified as 2S albumins [136], and then recorded as allergens due to their high immunoreactivity with sera from patients allergic to cashew nuts (>73%); these proteins were designated as Ana o 3 [136]. A high degree of identity and similarity was revealed between them and Pis v 1 (pistachio), Jug n 1 (black walnut), Jug r 1 (English walnut), and Car I 1 (pecan nuts), among others [129,137]. The classification as a major allergen [136] was confirmed by the reactivity of Ana o 3 with sera from 21 out of 26 patients with a clear history of moderate to severe allergic reactions to cashew ingestion [129].

#### 2.3.2. Immunochemical/Molecular Biology Analytical Methods for Cashew Detection

Several methods were developed for the specific detection of cashew proteins in food matrices. Lateral Flow Devices (LFD) and ELISA kits were applied to a broad spectrum of food matrices, with sensitivity of 1–2 or 0.2–1 mg/kg of cashews in food, respectively. However, some cross-reactivity occurred between cashews and other species, such as Brazil nut, pistachio, hazelnut, peanut, and walnut [138]. In addition to commercial kits, sandwich ELISA was applied to the detection of traces of cashew in different food matrices (wheat flour, rolled oats, milk chocolate, raisin bran cereal, chocolate-filled cookies, rice cereal) [29]. Furthermore, a multi-target indirect ELISA allowed the detection of cashews in milk and dark chocolate down to a 1 mg/kg level [139]. In a recent study by Zhao, Y. et al., two sensitive sandwich ELISAs specific for albumin 2S (Ana o 3) were developed as a stable protein marker for the detection of cashew residues in prepackaged food products. Development of ELISAs specific to Ana o 3 showed very low cross-reactivity with almond, pecan, pistachio, hazelnut, and peanut. The two assays were able to detect traces of cashew nut residues in processed chocolate and cookie [140]. LODs and LOQs achieved are shown in Table 1 along with other studies reporting the use of ELISA tests for cashew detection.

Real-time PCR approaches for the detection of cashew nuts in raw and processed foods (chocolates, cookies, ice cream, spreads, and pesto) were also proposed [141,142]. Most of them aimed at the detection of single copy genes (Ana o 1, Ana o 2, or Ana o 3) [24], exhibiting high specificity for cashews, without cross-reactivity with other plant and animal species. Multi-target methods were also proposed, enabling the detection of cashews in several food matrices with high specificity and sensitivity [35,36].

#### 2.3.3. Mass Spectrometry-Based Analytical Methods for Cashew Analysis

So far, few MS-based methods have been described in literature for the detection and quantification of cashew nut in food. In the LC-MS/MS multi-allergen method developed by Bignardi and co-workers in 2010, cashew detection in several food matrices (chocolates, biscuits, cakes, and flours) was investigated as well [91]. After the optimization of the sample protocol and the LC–ESI-LIT-MS/MS conditions for the simultaneous analysis of five nut protein allergens, including Ana o 2 of cashew, in cereals and biscuits, the sensitivity of the method was evaluated for all allergens investigated. In particular, LOD values ranging from 14 to 50 mg/kg matrix were obtained for nuts by working in SRM acquisition mode, with more sensitive LOD (14 µg/g) achieved for cashew. Such limits were generally higher than those reported for ELISA (0.2–1 mg/kg) or sandwich ELISA (0.1–1 mg/kg) kits; moreover, LOD and LOQ values were considerably higher when targeting cashew nuts in chocolate, compared to biscuits [143].

As a part of a multi-allergen method based on LC-HRMS equipped by a linear ion trap-Orbitrap mass spectrometer, new cashew protein peptides reported in Table 2 were proposed as markers by Korte et al. in 2016 [50]. Three marker peptides were detected even in the most diluted sample providing an LOD value of 2 mg/kg of protein in ice cream [50]. More recently three peptides from cashew proteins were identified in 2019 by Korte et al. in a general investigation on the impact of food matrices on the detectability of food allergen marker peptides [144]. Overall, it was shown that the kinetics of tryptic peptide formation, the degradation and the potential interference with the specific food matrix can influence the quantification of allergens and should be considered as a criterion for marker selection [144]. For example, heat treatment was shown to drastically affect allergen detectability, leading to a signal loss of 20–83% after baking bread and biscuits, depending on the allergen studied and the cooking time/type of food [144].

In the UHPLC-MS/MS method proposed by Planque et al. in 2017, cashew was detected in complex (chocolate, ice cream) and processed (cookie, sauce) foodstuffs achieving LOD value of 2.5 mgPROT/kg [20]. On the other hand, by exploiting a hybrid high-resolution mass spectrometer, namely a quadrupole-Orbitrap instrument, Gu et al. reported LOD and LOQ values of 0.7 µg/g and 2.3 µg/g of cashew, respectively, in chocolate, using the tryptic peptide ADIYTPEVGR from protein Ana o 2 as marker [92].

Very recently, New et al. proposed an LC-MS/MS-based method able to simultaneously detect 12 allergens, including cashew among the tree nuts species investigated. After selecting the most sensitive peptide for each allergen and evaluating the most important parameters for method validation, they found that the method was able to screen and identify 12 allergens at detection limit of 10 µg/g in incurred cookie and bread, including cashew [4]. Table 3 provides an overview of the most recent LC-MS methods developed for the quantification of nuts and peanuts in food.

### 2.4. Pistachio

#### 2.4.1. General Information and Main Allergens

Pistachio (*Pistacia vera*) belongs to the family of *Anacardiaceae* and althouh native to Central and Western Asian countries. It is widely cultivated also in the Americas, Europe, and Africa [145]. In 2018, the highest consumption of pistachios was reported in the United States (127,434 tons), followed by China (111,500 tons), Turkey (102,800 tons), and the European Union (101,800 tons) [146].

The widespread consumption of pistachio is mainly attributable to the benefits for human health and for the nutritional (Figure 5) and organoleptic properties. Similarly to what described for other nuts, pistachios shows a high level of phytosterols, which can contribute to a cholesterol-lowering effect; additionally, they have been connected to antioxidant properties [122,147]. Moreover, pistachios have also been used in traditional medicine to treat various clinical conditions, such as cirrhosis, abdominal discomfort, abscess, amenorrhea, bruises, sores, trauma, and dysentery [148,149,150,151,152].

On the other hand, the ingestion of pistachios can cause the onset of allergic reactions, triggering anaphylaxis with even fatal effects [153,154]. Currently, pistachio allergy is also under investigation for its relationship with cashew allergy. Five pistachio proteins (Pis v 1, Pis v 2, Pis v 3, Pis v 4 and Pis v 5) were officially defined as food allergens by the World Health Organization [145,155]. Among them, Pis v 4 was recognized as an iron/manganese superoxide dismutase, whereas the other four allergens belong to the group of seed storage proteins, namely Pis v 1 (2S albumin), Pis v 3 (7S globulins or vicilins), Pis v 2, and Pis v 5 (11S globulins or legumins) [145,155]. As previously mentioned, pistachios and cashews, belonging to the same family, share a similar protein expression profile and the expected cross-reactivity of IgE binding proteins has been confirmed by several studies [156,157,158,159].

In fact, the Pis v 1.0101 isoform of Pis v 1 shares part of its sequence with Ana o 3, a 2S cashew albumin. In particular, three tryptic peptides (ECCQELQEVDR, CQNLEQMVR, and ELYETASELPR) of Pis v 1 were found to be homologous to peptides of Ana o 3 from cashew [145]. Pis v 2 has 2 isoforms, Pis v 2.0101, and Pis v 2.0201, that show 48% and 46% identity, respectively, compared to the cashew legumin Ana o 2. Pis v 5, more closely related to Ana o 2 (79% amino sequence identity), shares the amino acid sequence at 52% and 51%, respectively, with proteins Pis v 2.0101 and Pis v 2.0201. Pis v 3 shares 80% amino acid sequence with cashew vicilin, Ana o 1, as well as a common sequential IgE epitope [132], which explains the high IgE cross-reactivity observed between these two proteins [133]. Rouge et al. as confirmation of the high homology between Pis v 3 and Ana o 1, identified an epitope in Pis v 3 (DEEQEEEDENPYVFED) that is almost identical to one epitope in Ana o 1 (DEAEEEDENPYVFED). Pis v 4 instead showed an 88% identity to manganese superoxide dismutases from latex, suggesting a further potential IgE cross-reactivity, but further studies on clinical relevance are needed [160].

#### 2.4.2. Immunochemical/Molecular Biology Analytical Methods for Pistachio Detection

Although not extensively investigated yet, some commercial polyclonal antibody (pAb)-based ELISA kits are available on the market for pistachio analysis [145]. In order to overcome the disadvantages of pAb-based assays, such as measurable cross-reactivity or the occurrence of false positive/negative results due to the impact of heat treatments on signal reduction/elimination, a murine mAb-based direct sandwich ELISA kit for pistachio detection has been marketed and tested by Liu et al. for its specificity, sensitivity, and robustness [161]. The test was found sensitive (limit of detection = 0.09 ± 0.02 ppm full fat pistachio), reproducible (intra- and inter-assay variability <24% CV), and rapid (post-extraction testing time ∼1.5 h). Moreover, the target antigen was found stable and detectable in pistachios subjected to different treatment, and no cross-reactivity was discovered in 156 food matrices tested, suggesting that this ELISA test is pistachio-specific. The most recent studies exploiting ELISA for cashew detection are summarized in Table 1.

As for PCR methods, detection limit of 0.1 and 4 mg/kg in wheat flour and biscuits, respectively, were obtained by amplification of multi-copy genes or region of pistachio genome. Multiplex PCR systems were found as effective tools for multiple imaging/quantification of allergenic products, including pistachio, in processed foods [106,162]. Concerning this aspect, Köppel at el. proposed two hexaplex real-time PCR systems for the simultaneous detection of 12 allergenic products, with LOD values for pistachio being 32 mg/kg in boiled sausages and 5 mg/kg in rice biscuits [35].

#### 2.4.3. Mass Spectrometry-Based Analytical Methods for Pistachio Analysis

Regarding the application of MS-based platform to the detection of pistachio allergens in foods, to date four methods aimed at identifying multiple allergens (including pistachio) in different complex matrices have been described [50,52,92,144,163]. In the first case, Sealey-Voyksner and co-workers reported a multiplex approach based on LC-MS/MS relying on a Q-TOF mass spectrometer to specifically detect 12 allergenic ingredients namely peanut, pecan, almond, cashew, hazelnut, walnut, Brazil nut, pine nut, pistachio, macadamia, coconut, and chestnut. After identifying marker peptides for allergens unique identification, method sensitivity was evaluated and an LOD of 1 mg/kg for all target allergens, regardless of food matrices kind investigated (cakes, biscuits, cereal bars, or chocolates) was reported. The same year Korte et al. described in their study the development of an LC/MS-based method for peanut and nuts detection and quantification in three food matrices by operating in MRM3 acquisition mode [164]. As for pistachio, the method targeted three peptides of the Pis v 5 allergen enabling the detection of 1 mg/kg of pistachio in vanilla ice cream, fortified multigrain bread, and milk chocolate. As a part of the application of an LC-HRMS method based on a linear ion trap-Orbitrap mass spectrometer, the same authors achieved LOD values ranging between 5 to 10 mg/kg, depending on the selected marker peptide [50]. Pistachio nut was investigated in the multiplexing method of Planque et al. for the simultaneous detection of ten allergens in complex foods such as ice cream and chocolate, and processed foods, namely sauce and biscuit. By calculating LOD as signal-to-noise ratio greater than 10 for the most abundant transition arising from MRM, Planque et al. detected pistachio allergens down to 2.5 mgprot/kg allergens in chocolate, ice cream, biscuit, and sauce [165]. Finally, Gu et al. proposed an LC-MS/MS method able to identify pistachio traces in chocolate, relying on the EGQLVVVPQNFAVVK peptides from the Pis v 2 protein, achieving an LOD of 0.4 mg/kg [92]. A summary of marker peptides adopted so far for the identification of pistachio traces in different food products is reported in Table 2. Table 3 explains an overview of the most recent LC-MS methods developed for the quantification of nuts and peanuts in food.

### 2.5. Walnut

#### 2.5.1. General Information and Main Allergens

Walnut, also known as common, Persian, English, Californian, or Carpathian walnut designates the species *Juglans regia*, which belongs to the botanical family of *Juglandaceae*, encompassing 24 different species. Among them, walnut is the most well-known species and, although native of Balkans region and different areas of Asia, it is currently widespread all over the world and prized for its nutritional (Figure 6), healthful, and sensory attributes. Walnut kernels represent a food with high nutritional value due to their fat content and richness in protein, vitamin, and mineral profiles; they also contain a wide variety of flavonoids, phenolic acids, and related polyphenols. Regular consumption of walnuts can reduce total plasma and low-density lipoprotein (LDL) cholesterol and have a positive effect on blood high-density lipoprotein (HDL) cholesterol and apolipoprotein A1 [166,167]. Walnuts are composed by a perfectly balanced n-6 and n-3 polyunsaturated fatty acids (present in an approximate ratio of 4:1), which has been related to decreased cardiovascular risk incidence [168]. Other benefits related to their intake include the reduction of inflammation and the improvement of arterial function [92,169,170].

However, as for other major tree nuts, walnut ingestion may pose a health risk due to the possibility of inducing hypersensitivity in sensitized/allergic individuals. In fact, one of the most important nut allergies in Europe is linked to the consumption of walnut [171]. Since 1993, walnut and other tree nuts have been defined as one of the 8 groups responsible for almost 90% of human food allergies. According to EU legislation, walnut falls into one of 14 food classes and substances that cause allergies or intolerances and which must be emphasized among ingredients listed in processed foods, regardless of their quantity [172]. Several allergenic proteins have been identified and characterized in walnut. The most important belong to the prolamine (Jug r 1 and Jug r 3) and cupin (Jug r 2 and Jug r 4) superfamilies of proteins. More recently, Jug r 5 (a PR-10 protein), Jug r 6 (a vicilin-like cupin), Jug r 7 (a profilin), and Jug r 8 (a ns-LTP-2 protein) have also been included in the Allergome database (www.allergome.org, accessed on 10 July 2021). Jug r 1 allergen is considered of great clinical importance because it has been associated to severe allergic symptoms in walnut-allergic patients [173]. 2S Albumins, the class of proteins to which Jug r 1 belongs, reveal high resistance to thermal denaturation and to enzymatic activity (trypsin/chymotrypsin) at basic pH (8.0), although they progressively lose their allergenicity under acidic conditions.(pH 1.3) in the presence of pepsin [174]. Included in the group of nsLTP, the Jug r 3 protein is associated with severe and systemic allergic reactions that can be potentially life-threatening [175,176]. The effect of food processing on Jug r 3 allergen is not yet clear, although heat treatments above 90 °C for long times (15 to 30 min depending on temperature) appear to reduce the allergenicity of some nsLTPs of other species [177]. Jug r 2 protein is a vicilin identified as an important allergen as well. After submitting nuts to single or combined processes, including exposure to gamma rays, microwaving, roasting, frying, blanching and/or autoclaving, Su et al. found a good stability of Jug r 2 protein which retained its allergenicity in all cases [178]. The same outcome was obtained for Jug r 4, an 11S globulin (legume type) that is also considered an important allergen of the common walnut [80,178]. Finally, limited information is available on the profilin Jug r 5, likely responsible for mild clinical symptoms mostly occurring after raw food ingestion and often confined to the oral cavity [175].

#### 2.5.2. Immunochemical/Molecular Biology Analytical Methods for Walnut Detection

Several ELISA kits are commercially available for detecting walnut proteins down to 0.25–0.35 mg/kg in various food matrices, such as biscuits, ice cream, or chocolate [172]. However, some of them suffer of cross-reactivity with multiple foods: (i) nuts belonging to the same walnut botanical family (*Juglandaceae*), (ii) other tree nuts (pistachio, hazelnut, Brazil nut, chestnut, pine nut), and (iii) food belonging to other plant species (quinoa, sesame, buckwheat, and soybean) [172]. Besides the commercial kits, non-competitive sandwich-type ELISA and indirect competitive ELISA have been proposed in the literature to trace walnut allergens in foods with sensitive LOD values (<1 mg/kg of walnut protein) [179,180,181,182].

In 2021 Madrid et al. have carried out a survey of commercial food products for detection of walnut by using three different methods: a sandwich ELISA kit based on polyclonal antibodies, a direct ELISA based on recombinant multimeric scFv, and a real time PCR. Respectively, the first ELISA was less influenced by sample processing than was the second one, but cross-reactivity with pecan was recorded, producing some false positives that need to be confirmed by real time PCR. In the samples analyzed, walnut was present in 7.0–12.6% of foods that did not declare their presence, confirming the risk for allergic consumers [183]. LODs and LOQs values are shown in Table 1 together with the most recent studies exploiting ELISA for walnut detection are reported.

Walnut accidental contamination was investigated also by molecular approach. Among the DNA-based methods, several PCR assays have been reported in the literature for the specific detection of walnut in processed foods. Two commercial real-time PCR kits are available for the qualitative analysis of nuts in food: the SureFood real-time PCR kit, for which very low LOD and LOQ values are indicated (0.4 and 10 mg/kg, respectively), and the NutsKit real-time PCR, developed to target both walnut and pecan nut, presenting no cross-reactivity with twenty other allergens. Real-time PCR methods for walnut detection are based on the amplification of a region encoding for the allergenic proteins Jug r 2 or Jug r 3 [35,184,185,186,187]. Almost all the proposed systems can be virtually applied to a wide variety of processed foods (e.g., cakes, biscuits, sausages), allowing for the quantification of walnuts down to 5–10 mg/kg or a few copies of DNA.

#### 2.5.3. Mass Spectrometry-Based Analytical Methods for Walnut Analysis

Similarly to what was described for other nuts, walnut was included in a number of multi-target MS-based method for the detection of allergen in raw and complex food [49,91,92,163,164,188]. Specifically, in their multitarget LC-ESI-LIT-MS/MS-MS3 method for the analysis of biscuits samples fortified with a mix of peanut and nuts (hazelnuts, walnuts, almonds, and cashews) Bignardi et al. obtained an LOD of 55 or 50 mg/kg for the most sensitive LDALEPTNR (Jug r 4) peptide marker by working in MS/MS or MS3 acquisition mode, respectively [91]. Further optimization of the analytical protocol already published in 2010 led the same authors to obtain much lower LOD values for walnut, namely 0.8 mg/kg by detecting the ADIYTEEAGR (Jug r 4) peptide in fortified biscuits, and 5 mg/kg using the LDALEPTNR (Jug r 4) peptide in fortified dark chocolate [143].

In the study by Heick et al. above cited, LOD of 70 mg/kg for walnut in bread using the DLPNECGISSQR (Jug r 1) peptide was achieved [49]. Several further walnut marker peptides have been selected over the years by other authors, as reported in Table 2. In a study carried out by Sealey-Voyksneret al., selected peptides were screened for walnut taking into account potential problems with stability, including hydrolysis of aspartic acid, oxidation of methionine and deamidation of glutamine under basic conditions [163].

More recently Korte et al. reported an LC-HR-MS/MS based method able to obtain LOD values ranging from 5.7 to 35.7 mg/kg for walnut detection [50] lowered to below 1 mg/kg for nuts in bread and ice cream when the MRM acquisition mode based on MS3 measurements was implemented [164]. On the other hand, the UHPLC-MS/MS-based method developed by Planque et al. in 2017 for multi-allergens detection in foods led to detect down to 5mg/kg of walnut in processed cookies, tomato sauce (kept at 95 °C for 45 min), chocolate, and ice cream [52]. Finally, by selecting a Jug r 2 tryptic peptide VFSNDILVAALNTPR as a marker for walnut detection in incurred chocolate, LOD down to 2 mg/kg level was reported in the study carried out by Gu et al. [92]. In a comparative study on the kinetic of release of marker peptides from the corresponding protein allergens, also including the assessment of the interference of the matrix with peptide detection and the estimation of heat treatments effect on the recovery of allergens, the behavior of walnut protein tryptic peptides FFDQQEQR (Jug r 2), ATLTLVSQETR (Jug r 2), and ALPEEVLATAFQIPR (Jug r 4) were studied. In particular, the recovery of walnut proteins was found to increase when baked goods were analyzed, despite what was observed for chocolate matrix, which was found to affect the signal intensity of marker peptides leading to their decrease. Additionally, heat treatment-affected allergen detectability, leading to a loss of signal after baking bread and cookies [144]. The most relevant peptides reported as markers for the detection/quantification of walnut allergenic proteins in food products are described in Table 2. Table 3 shows an overview of the most recent LC-MS methods developed for the quantification of nuts and peanuts in food.

### 2.6. Peanut

#### 2.6.1. General Information and Main Allergens

Peanut (*Arachis hypogaea* L.) belongs to the *Fabaceae* or *Leguminosae* family, native to South America [165]. Currently, peanuts are cultivated in China, India, Africa, Japan, South America, and the United States, with more than 300 varieties distributed worldwide [165]. Peanut has great economic and nutritional value (see Figure 7 for more details) and is widely used in the food industry to produce peanut butter, seed oil, snacks, soups, desserts, and for direct consumption [189]. Since peanuts are technically legumes, they are richer in protein and more nutritionally complete than nuts [122]. Peanut seeds contain approximately 22–30% of crude protein [165] and are an excellent vegetarian source of protein and healthy fats. In past years, peanuts, like tree nuts, have often been perceived as an unhealthy food due to their high fat content estimated to be ≥50% *w*/*w* [190]. In contrast with this, several studies have shown that regular peanut consumption is linked to a reduction in the incidence of heart disease [191] and some tumors [192]. Moreover, an improvement in weight management [193] was observed. Additionally, other studies have demonstrated the richness of peanuts in phytonutrients with high nutritional value which contribute to improve the overall human health and well-being [194]. Among macronutrients, a high content of starch carbohydrate was found in dry roasted peanuts [189], while water-soluble B vitamins (involved in vital reactions in energy metabolism, cholesterol synthesis and heme and DNA synthesis) and fat-soluble vitamin E (tocopherol, with important antioxidant activity) [165] were observed in the peanut micronutrient fraction. Finally, peanuts are rich in macrominerals, such as potassium, phosphorus, and magnesium, which are important from a biological point of view for electrolyte balance, hydration, and proper functioning of nerves and muscles [165] and trace elements, such as zinc, iron, copper, and selenium, involved in various biological functions. Despite the health benefits associated with its consumption, peanut is a leading cause of fatal and near-fatal anaphylaxis [195] because it is counted among the major foods able to trigger allergy [196]. Allergic reactions to peanuts are often severe, can occur upon first ingestion, and commonly become more severe over time or after repeated exposure [197].

To date, 18 peanut proteins (Ara h 1 to Ara h 18) have been identified as responsible for peanut allergy by the World Health Organization and International Union of Immunological Societies Allergen Nomenclature Sub-Committee (WHO/IUIS, 2021). They have been associated to seven different protein families: cupins (Ara h 1, Ara h 3), 2S albumins (Ara h 2, Ara h 6, Ara h 7), profilin (Ara h 5), PR -10 (Ara h 8), nsLTP type 1 and type 2 (Ara h 9, Ara h 16, Ara h 17), oleosins (Ara h 10, Ara h 11, Ara h 14, Ara h 15), and defensins (Ara h 12, Ara h 13) [165,198]. Ara h 1, Ara h 2, Ara h 3, and Ara h 6 have been reported as the major allergens [199,200]. Specifically, Ara h 6, stable to heat and digestion, showed a similarity of 60% of sequence identity with Ara h 2, and it was reported to be recognized by 30–40% of patients’ IgE antibodies [201,202]. It is important to remember that peanut allergens have shown extreme resistance to proteolytic digestion and to thermal or chemical denaturation [203,204].

#### 2.6.2. Immunochemical/Molecular Biology Analytical Methods for Peanuts Detection

Peanuts are a typical example of allergenic ingredient that can pose a relevant risk for allergic consumers if present in raw materials, semi-finished, or finished food products that should not contain them. In a European study, peanuts were found in 25% of biscuits and 43% of chocolate whose labels included a precautionary phrase indicating their possible presence as contaminants [205]. Unfortunately, traces of peanuts were found also on 11% of biscuits and 25% of chocolate products that did not have warning phrases on their labels [12,205].

Therefore, the importance of developing effective analytical methods that can detect traces peanuts is evident. As far as immunochemical methods are concerned, Poms et al. performed an interlaboratory validation study of five ELISA test kits, searching for traces of peanut in biscuits and dark chocolate [206]. They demonstrated good reproducibility, but false negatives were observed in as many as 25% of the dark chocolate samples for some kits. Montserrat et al. in 2015 developed sandwich and competitive ELISAs for Ara h 1 and Ara h 2 to detect peanut in biscuits prepared with peanut butter. Competitive format showed greater sensitivity than sandwich format for both proteins. The sandwich format for Ara h 2 is able to detect the addition of 2.5% peanut butter, instead the same format for Ara h 1 could not detect 5% added peanut. Direct competitive ELISA could detect the peanut butter addition of Ara h 1 and Ara h 2 of 1% and 0.05%, respectively [207]. Table 1 shows LODs and LOQs achieved. In a 2020 study, a commercial ELISA (R-Biopharm) was used to perform an investigation that determined the extent of undeclared allergens in food products imported to the Asian retail market in Australia. ELISA kits used for the analysis of soft drinks, pastry, bread, and baked goods, mixed and/or processed foods achieved an LOD of 0.13 mg/kg and an LOQ of 2.5 mg/kg for peanuts proteins [208]. A similar study was conducted in 2021: samples were analyzed for peanuts (*n* = 871) using r-Biopharm’s sandwich ELISA kits, RIDASCREEN^®^FAST Peanut. Within the samples analyzed, 72% had a PAL (*n* = 628), 1% had peanuts as a minor ingredient (*n* = 9), and 27% were declared “peanut-free” (*n* = 234). Peanuts were found in 4% (0.6–28.1 ppm) of all samples. The kit used is characterized by an LOD > 0.03 mg/kg and an LOQ of 0.555 mg/kg as shown in Table 2 [107].

Recently, DNA-based detection methods have been introduced to detect also peanut proteins in food [34,35,36,209]. Several real-time PCR tests have been also developed [210,211,212], with the potential to be applied for quantitative purposes in other comparative studies between PCR and ELISA technologies designed to detect traces of peanuts in food products [211,213]. Scaravelli et al. [212], shown that despite the completely different nature of the target molecules, proteins or DNA, the performance of the two detection methods was very similar.

#### 2.6.3. Mass Spectrometry-Based Analytical Methods for Peanuts Analysis

Several LC-MS/MS methods based on the use of ion-trap, triple-quadrupole or QTOF MS are reported in the literature used to identify markers in raw or roasted peanut or detect traces of peanut in complex foods [42]. Specifically, Table 2 collects peptide markers selected in the literature over the years to recognize and quantify peanuts in different matrices using MS-based techniques. Different commodities have been investigated for the development of suitable methods for tracing peanuts, such as biscuits, chocolate, and ice creams, with variable LOD values obtained for each of them [47,49,214,215]. In ice cream model matrix, the detection of some specific peptides arising from tryptic digestion allowed to uniquely identify Ara h 1 and quantify it down to 10 mg/kg concentrations [47]. Peptide characterization was accomplished by exploiting a quadrupole-time of flight system combined with peptide sequence tag analysis and database search [47]. Successively, a comparative investigation aimed at evaluating the performance of two different LC-MS methods based on two different mass analyzers, namely a single quadrupole spectrometer and a triple quadrupole system, was accomplished by the same authors for detecting peanut in chocolate [214]. Interesting results were obtained by Chassaigne and co-workers by using a more sophisticated LC-MS instrumentation, namely capillary column liquid chromatography coupled with a quadrupole- time of flight mass spectrometer equipped with a nano-electrospray interface (CapLC-nano-ESI-Q-TOF MS/MS), to detect peptides arising from peanut allergenic proteins Ara h 1, Ara h 2, and Ara h 3 after enzymatic digestion [42]. The detection limits for single peptides, reported as absolute amounts injected in column, were 7 ng for unroasted peanuts, 10 ng for medium roasted peanuts, and 40 ng for highly roasted peanuts [42]. In 2017 Pilolli et al. developed an HPLC-SRM method coupled to the on-line enrichment by C18 solid phase extraction of tryptic digests to detect peanut in incurred cookies down to 13 µg/g [113]. As an alternative, the same authors developed a high-resolution MS-based method working properly on the same matrix and with promising results for tracing low levels of peanuts in cookie [114]. In a recent study by Zhang et al., a robust UPLC-Q-TOF method was developed for the quantification of peanut allergens in baked foods, based on signature peptides of the major peanut allergens Ara h 1 and Ara h 2 [216]. After optimizing the digestion process and investigating on the candidate peptides, DLAFPGSGEQVEK and NLPQQCGLR were selected as the signature peptides for Ara h 1 and Ara h 2 detection, respectively. The corresponding isotopically labelled peptides were synthesized and applied to sample analysis as internal standards and limits of quantification of 0.30 and 0.13 mg/kg were calculated for Ara h 1 and Ara h 2, respectively [216]. Very recently, a comparative study between ELISA and LC-MS/MS based methods for the detection of peanut in wheat flour-based dry matrices was described by Chang [217]. In the light of an overestimation of peanut level observed by analyzing the target matrices by ELISA approach, the authors developed a simple and sensitive method based on LC-MS/MS technology to overcome immunological test disadvantage. After identifying the proper marker peptides for quantitative analysis, namely DLAFPGSGEQVEK and IFLAGDKDNVIDQIEK, arising from Ara h 1 digestion, the authors validated the method taking into account parameters such as linearity, sensitivity, recovery, and stability. Interestingly, linearity was tested by using the protein Ara h 1 as standard compound to prepare matrix-matched calibration curves after dilution in matrix extract. This strategy enables to provide results expresses as Ara h 1 (mg/kg) or total peanut (mg/kg) by exploiting a simple conversion factor (28.6*x* in this method). The quantification of the method was reported to be comprised between 0.31 and 40 mg/kg total peanut, while variable recovery values ranged from 51.1% to 104.2% for peptide DLAFPGSGEQVEK and from 54.4 to 113.0% for peptide IFLAGDKDNVIDQIEK were obtained depending on the wheat-based matrix considered [217]. In the recent years, high-resolution tandem MS based on Orbitrap mass analyzer for multiple analysis of allergen proteins in food products, including those from peanuts, have been reported [20,50,92,112,114,218,219]. By exploiting a stand-alone Obitrap-based analyzer, Monaci and colleagues detected and quantified traces of peanut proteins in a flour obtained from a mixture of nuts (hazelnuts, pistachios, almonds and walnuts). By following the peptides VYDEELQEGHVLVVPQNFAVAGK and WLGLSAEYGNLYR, detection limit of 4 μgprotein/g were achieved [220]. Furthermore, marker peptides such as GTGNLELVAVR (Ara h 1) and RPFYSNAPQEIFIQQGR (Ara h 3/4) allowed to detect peanut down to LOD values of 0.8 mg/kg and 1.3 mg/kg and LOQ values of 2.6 mg/kg and 4.3 mg/kg in chocolates spiked with allergenic ingredients [92]. In the same year, Boo et al. detected up to 5 mg of peanuts per kg of sugar cookies (cooked for 25 min at 190 °C) [219] while an LOD of 2.5 mg/kg was obtained for peanut selected peptides in processed biscuits, tomato sauce (treated for 45 min at 95 °C), chocolate, and ice cream [52]. In a recent study, more than 300 peptides were identified during the analysis of various processed peanut matrices (i.e., raw peanuts, heated peanuts, a low-pH peanut matrix, caramelized peanuts, and peanuts in a fat-rich environment). A careful study of the nature of these peptides (multiple protein isoforms and origin variation issues, abundance, resistance to heat treatment, or chemical modifications) was carried out to ensure UHPLC-MS/MS method specificity, sensitivity, trueness, and robustness. This approach led to the selection of 16 potential peanut peptide biomarkers reported in Table 2 [221]. In a parallel study, the same authors presented and compared the performances of three different types of isotopically labelled internal standards to quantify allergens in processed food products (cookie, chocolate, and unbaked lyophilized cookie dough): synthetic peptides, concatemer, and protein. However, isotopically labelled synthetic peptides do not exactly reflect the natural situation as they do not need to undergo proteolytic digestion on which much of the observed variability could depend. Concatemers need to be digested to release their constituting peptides. In the future perspective, isotopically labeled concatemers could represent relevant internal standards, as they combine the advantages of using labeled proteins overcoming the limitations of the use of synthetic peptides, and, in addition, allow the multiple quantification of allergens by MS. Peptides of the allergenic proteins of interest used in this study are shown in Table 2 [222]. Along with other major food allergens, peanut was included as target molecule in the LC-MS/MS based method developed by New et al. in 2020 for multiple allergens detection in selected food and accepted as First Action by the AOAC Official Method Boards as in compliance with the Standard Method Performance Requirements (SMPR) 2016.002. The method was validated for cookies and breakfast cereal matrices and an MDL of 2.2 and 1.2 mg/kg were respectively estimated in the case of peanut. Table 3 shows an overview of the most recent LC-MS methods developed for the quantification of nuts and peanuts in food.

**Table 1 foods-11-00728-t001:** Overview of the most recent ELISA kit developed for the detection of peanut and nuts allergens reported in literature.

Allergen	Description	Target Protein	Matrix	LOD	LOQ	Cross-Reactivity	Reference
ALMOND	Sandwich 4C10 ELISA	Pru du 6	Biscuit, Brittle, Granola bar with raw almond, Granola bar with roasted almond	0.97 ± 0.32 mg/kg full fat almond	3.24 ± 1.07 mg/kg full fat almond	Not reported	[87]
	Sandwich MonoTrace ELISA			0.30 ± 0.17 mg/kg full fat almond	1.01 ± 0.58 mg/kg full fat almond		
	Sandwich	Pru du 6	Vanilla ice cream, cookies, pasta sauce, milk chocolate	0.3 mg/kg	2.5 mg/kg	No cross-reactivity	[88]
	Sandwich	Pru du 6	Orange juice, Coffee liquor, chocolate soy drink, salad dressing, rice ice cream, goat cheese	0.02 mg/kg	0.12 mg/kg	Low cross reactivity to chestnut	[90]
HAZELNUT	Sandwich	Hazelnut proteins	Cookies, chocolate	>0.03 mg/kg	0.375 mg/kg	Not reported	[107]
	Sandwich-Ridascreen Fast Hazelnut	Hazelnut proteins	31 plant-derived foods and 2 animal-derived foods	1.5 mg/kg	2.5 mg/kg	No apparent cross-reactivity	[103]
	Sandwich-AgraQuant Hazelnut Assay	Hazelnut proteins	30 plant-derived foods	0.3 mg/kg	1 mg/kg	No cross-reactivity	
	Sandwich-DIA hazelnut	Hazelnut proteins	31 plant-derived foods	0.33 mg/kg	1 mg/kg	No cross-reactivity	
	Sandwich-Veratox for hazelnut	Hazelnut proteins		2.5 mg/kg	2.5 mg/kg	No available information about specificity	
	Sandwich	Protein from ground hazelnut & hazelnut chocolates	Dried hazelnuts and chocolate with 41% cocoa	1 mg/kg	50–100 mg/kg	Not reported	[37]
CASHEW	Sandwich (CAS-ELISA-2)	Ana o 3	Chocolate	0.04 mg/kg	0.4 mg/kg	Very low cross-reactivity with pistachio, pecan, almond, peanut, and hazelnut.	[140]
			Cookie		40 mg/kg		
	Sandwich (Ano3-ELISA-1)	Ana o 3	Chocolate	0.06 mg/kg	0.4 mg/kg		
			Cookie		40 mg/kg		
	Sandwich	Cashew proteins	Ice Cream, cookies, chocolate, nuts/ Mixed nuts, breakfast/granola or Nutritional bars, nut butter	0.11 mg/g	0.46 mgcashew g_1	Not reported	[223]
PISTACHIO	Sandwich enzyme-linked immunosorbent assay (ELISA)	Pistachio proteins	156 commonly used foods and food ingredients	0.09 mg/kg	0.30 mg/kg	No cross-reactivity	[161]
	Sandwich ELISA AgraQuant^®^ Plus Pistachio	Pistachio proteins		1 mg/kg	1–25 mg/kg	Cashew (12%), hazelnut (0.17%), walnut (0.0008%), pecan nut (0.0005%), sunflower (0.0002%)	[145]
	Sandwich-AgraQuant^®^ ELISA Pistachio	Pistachio proteins		0.13 mg/kg	1–40 mg/kg	Cashew (12%), hazelnut (0.17%), walnut (0.0008%), pecan nut (0.0005%), sunflower (0.0002%)	
	Monoclonal antibody-based ELISA to pistachio allergen	Pistachio proteins		0.12 mg/kg	1–40 mg/kg	Pecan nut (0.001%)	
WALNUT	Indirect competitive	Jug r 1	Tree nuts, seeds, cereals, soy, milk, various animal products	0.22 mg/kg walnut protein	0.44 mg/kg walnut protein	0.2% to pecan	[182]
	Sandwich ELISA	Walnut proteins	Sauce, beverage, yoghurts, ice-cream, and sandwich	2.2 mg/kg	3.3 mg/kg	Cross-reactivity to almond and pecan	[183]
	Direct ELISA with Multimeric scFv			1616 mg/kg		Cross-reactivity was found with pecan (2.25%) and almond (0.35%)	
	Sandwich	Jug r 1	chicken meatballs, rice porridge, bread, sponge cake, orange juice, jelly, biscuit	<0.16 mg/kg of walnut protein defined by the calibration curve	<0.31 mg/kg	Strong cross-reactivity with pecan nut, hazelnut, brazil nut, almond, pine nut, peanut, cashew nut, pistachio, macadamia and mustard	[180]
PEANUT	Sandwich-	Ara h 1 Ara h 2	Biscuits prepared with peanut butter	Ara h 1: 0.1 mg/kg Ara h 2: 0.13 mg/kg	20–800 mg/mL	Slight interference nuts, seeds, cereals	[207]
	Direct competitive			Ara h 1: 0.19 mg/kg Ara h 2: 0.06 mg/kg	20 ng/mL–2 mg/mL	Slight with nuts, seeds, cereals	
	Sandwich	Ara h 2	Peanut kernels, peanut beans, peanut crispy rolls, chocolate-peanut beans	0.7–1.7 μg/kg of peanut product	No information	Low cross-reactivity to cashews, walnuts, BSA, ovalbumin, soy & pea proteins	[224]
	Sandwich	Peanut proteins	Cookies, chocolate	>0.03 mg/kg	0.555 mg/kg	Not reported	[107]
	Sandwich	Peanut proteins	Non-alcoholic beverages, confectionery, bread and bakery, mixed and/or processed foods	0.13 mg/kg	2.5 mg/kg	Not reported	[208]

**Table 2 foods-11-00728-t002:** Collection of the main allergenic proteins of almond, hazelnut, pistachio, cashew, walnut, and peanut and of peptides exploited for their identification and quantifications as hidden allergens in food products using analytical approaches based on mass spectrometry.

ALMOND-*Prunus dulcis*				
Allergenic Protein	Molecular Weight	Food Processing Effects	Selected Peptide Sequences	References
Pru du 5 (Q8H2B9) (60s acidic ribosomal prot. P2)	10 kDa	Unknown	DITELIASGR	[225]
Pru du 6 (E3SH28; E3SH29) (Amandin, 11S globulin legumin-like protein)	360 kDa	Thermally stable to dry heat such as roasting but it can be denatured by boiling.	QETIALSSSQQR	[50,52,112,225,226]
			GNLDFVQPPR	[48,49,50,92,112,144,164,225,226,227,228,229]
			ALPDEVLANAYQISR	[48,49,92,112,225,226,227,228]
			ISTLNSHNLPILR	[163,225,226,227]
			NGLHLPSYSNAPQLIYIVQGR	[48,49,225,227]
			QQEQLQQER	[91,92,143,225,229]
			QQEQEQER	[229]
			TEENAFINTLAGR	[163,225,226,227]
			GVLGAVFSGCPETFEESQQSSQQGR	[48,225,227]
			VQGQLDFVSPFSR	[50,144,164,225]
			ALPDEVLQNAFR	[50,144,164,225,226,229]
			ADIFSPR	[225,226,229]
			LSQNIGDPSR	[225,229]
			VQVVNENGDPILNDEVR	[50,229]
			VQVVNENGDPILDDEVR	[225,229]
			NGIYSPHWNVNAHSVVYVIR	[225]
			NLQGQNDNR	[50,225]
			FYLAGNPENEFNQQGQSQPR	[225,226]
			QQGQQEQQQER	[91,226]
			QQEEQQSQR	[229]
			QQEQQQGQQGRPQQQQQFR	[225]
			QEGGQGQQQFQGEDQQDR	[229]
			FYLAGNPQDEFNPQQQGR	[225]
			NHLPILR	[225]
			ADFYNPQGGR	[92,225]
			LLSATSPPR	[50,92,225]
			QQQQQGEQGR	[229]
			NQIIQVR	[226]
			ENIGNPER	[226]
			TDENGFTNTLAGR	[50]
**HAZELNUT-*Corylus avellana***				
Cor a 8 (Q9ATH2) (Non-specific lipid transfer protein type 1)	9 kDa	Resistant to the activity of gastric and intestinal enzymes, heat treatments, abrupt changes in pH and the inhospitable environment of the gastrointestinal tract (proteolysis). Less stable when subjected to temperatures above 90 °C	GIAGLNPNLAAGLPGK	[37,109]
			AVNDASR	[110]
Cor a 9 (Q8W1C2) 11S seed storage globulin (legumin-like)	40 kDa	Thermostable protein, sensitive to autoclaving processes (121 °C or 138 °C, 15 or 30 min)	LNALEPTNR	[50,110,144,163,164,221,226,228]
			INTVNSNTLPVLR	[4,37,48,49,50,109,163,228,230]
			WLQLSAER	[37,92,109,163,226,230]
			ALPDDVLANAFQISR	[37,48,49,109,112,113,114,226,228,230]
			VQVVDDNGNTVFDDELR	[144,164,230]
			QGQQQFGQR	[230]
			EGLYVPHWNLNAHSVVYAIR	[110]
			ADIYTEQVGR	[4,47,48,49,66,90,91,111,112,141,214,220,225,226,229]
			QEWER	[91]
			AESEGFEWVAFK	[226]
			QETTLVR	[226]
			TNDNAQISPLAGR	[114,226,228]
			HFYLAGNPDDEHQR	[110,230]
			QGQVLTIPQNFAVAK	[37,48,49,50,109,112,114,144,164,228]
			INTVNSNTLPVLR	[226]
			QGQVLTIPQNFAVAK	[226]
Cor a 11 (Q8S4P9) 7S seed storage globulin (vicilin-like)	48 kDa	Thermostable proteins with major thermal transition of around 70–75 °C; The physico-chemical properties of Cor at 11 were affected after heat treatment at 60 °C and at 145 °C in the presence of glucose; High pressure processing (300–600 Mba) of the hazelnuts did not affect the ability to bind the IgE of the allergens of Cor a 11.	LLSGIENFR	[4,37,109,110,163,226,230]
			GNIVNEFER	[110]
			VQVLENFTK	[226]
			ALSQHEEGPPR	[230]
			HPSQSNQFGR	[230]
			ALSQHEEGPPR	[230]
			GSMAGPYYNSR	[230]
			IPAGTPVYMINR	[230]
			ESFNVEHGDIIR	[230]
			NQDQAFFFPGPNK	[230]
			IWPFGGESSGPINLLHK	[110]
			ILQPVSAPGHFEAFYGAGGEDPESFYR	[110,230]
			AFSWEVLEAALK	[4,37,109,226,230]
			ELAFNLPSR	[37,109,226,230]
Cor a 14 (D0PWG2) (2S albumin)	10 kDa reducing	High thermal stability and in difficult conditions of the gastrointestinal tract due to their compact and rigid structure, thus preserving their allergenic activity.	QAVMQQQGEMR	[230]
			QQNLNQCQR	[230]
**PISTACHIO–*Pistacia vera***				
Pis v 1 (B7P072) 2S Albumin	7 kDa	Unknown	LQELYETASELPR	[163,228]
			TNGLSQTSQLAGR	[163]
Pis v 2 (B7P073; B7P074) 11S Globulin subunit	32 kDa	Unknown	VTSINALNLPILR	[20,112]
			ALPLDVIK	[112,228]
			TNGLSQTSQLAGR	[50]
			GLPLDVIQNSFDISR	[50]
			NSFDISR	[92]
			EGQLVVVPQNFAVVK	[92]
			IQIVSENGESVFDEEIR	[50]
Pis v 3 (B4X640) Vicilin	55 kDa	Unknown	IAIVVSGEGR	[50]
			STGTFNLFK	[50]
Pis v 5 (B7SLJ1) 11S Globulin subunit	36 kDa (acidic subunit)	Unknown	ITSLNSLNLPILK	[112,144,164,228]
			AMISPLAGSTSVLR	[50,112,144,164,228]
			GFESEEESEYER	[50,144,164]
**CASHEW-*Anacardium occidentale***				
Ana o 1 (Q8L5L5; Q8L5L6) Vicilin-like protein	50 kDa	Resistant to pH and high temperatures. IgE reactivity was strongly reduced in cashew nut subjected to gamma irradiation at followed by autoclaving at 121 °C during 30 min.	AFSWEILEAALK	[50]
			CAGVALVR	[112,228]
			AMTSPLAGR	[92,112,143,228]
			ADIYTPEVGR	[50,91,92,143,144,163,228]
Ana o 2 (Q8GZP6) Legumin-like protein	55 kDa	Thermostable protein Ana or 2 immunoreactivity was markedly reduced with high sodium sulfite concentrations (≥50 mM) and high temperatures (≥100 °C).	LDALEPDNR	[92]
			WLQLSVEK	[50,134]
			TSVLGGMPEEVLANAFQISR	[50,134]
			EGQMLVVPQNFAVVK	[50,134,144]
			LTTLNSLNLPILK	[50,134,144]
			VFDGEVR	[91]
Ana o 3 (Q8H2B8) 2S albumin	14 kDa	High thermal stability Stable over a pH range of 1 to 11.	ELYETASELPR	[112,228]
			QLQQQEQIK	[163]
**WALNUT–*Juglans regia***				
Jug r 1 (P93198) 2S albumin seed storage protein	15–16 kDa	High resistance to enzymatic activity (trypsin/chymotrypsin) at basic (pH 8.0), although they progressively lose allergenicity in acidic conditions (pH 1.3) in the presence of pepsin. Resistance to thermal denaturation.	DLPNECGISSQR	[48,49,50]
			QCCQQLSQMDEQCQCEGLR	[48,49]
			GEEMEEMVQSAR	[48,49,112,163,228]
Jug r 2 (Q9SEW4) Vicilin seed storage protein	44 kDa	Remarkable thermal stability, which allows them to maintain their conformation at temperatures below 70–75 °C. However, when subjected to elevated temperatures, 7S globulins can undergo structural disruption and covalent modifications.	ATLTLVSQETR	[50,112,144,164,228]
			HNPYYFHSQSIR	[50]
			FFDQQEQR	[50,92,144,164]
			DFLAGQNNIINQLER	[92]
			VFSNDILVAALNTPR	[92]
			QQQQQGLR	[163]
Jug r 4 (Q2TPW5) 11S globulin seed storage protein	58.1 kDa	High thermal stability and great resistance to proteolysis, which allows them to maintain their allergenic properties along the gastrointestinal system.	ALPEEVLATAFQIPR	[50,144,164]
			EGQLLTIPQNFAVVKR	[50]
			LDALEPTNR	[91,143]
			NFYLAGNPDDEFR	[50]
			EFQQDR	[91]
			ADIYTEEAGR	[143]
**PEANUT-*Arachis hypogaea***				
Ara h 1 (P43238) Cupin (Vicillin-type, 7S globulin)	64 kDa	Heat stable protein that undergoes irreversible denaturation at T > 80 °C. Roasting at T > 140 °C produces an increase in the IgE binding capacity of Ara h 1. Hydration before autoclaving increases the effectiveness of the heat treatment by significantly altering its immunoreactivity.	DLAFPGSGEQVEK	[4,41,47,48,212,215,216,225,226,227,230,231,232]
			VLLEENAGGEQEER	[4,41,49,112,212,214,225,226,227,230,231,232]
			GTGNLELVAVR	[24,47,48,91,112,161,218,220,227,229,230]
			SFNLDEGHALR	[47,92,163,231]
			NNPFYFPSR	[25,47,92,219,229,233]
			NTLEAAFNAEFNEIR	[47,188,231]
			EGALMLPHFNSK	[231]
			ISMPVNTPGQFEDFFPASSR	[231]
			EEEEDEDEEEEGSNR	[231]
			EGEQEWGTPGSHVR	[216,231]
			AMVIVVVNK	[114,231]
			EHVEELTK	[231]
			IVQIEAKPNTLVLPK	[231]
			QFQNLQNHR	[231]
			EGEPDLSNNFGK	[231]
			DQSSYLQGFSR	[110,216,226,231]
			SSENNEGVIVK	[229,231]
			NNPFYFPSR	[221]
			GSEEEGDITNPINLR	[110,221]
			GSEEEDITNPINLR	[221,222]
			IFLAGDKDNVIDQIEK	[217]
Ara h 2 (Q6PSU2-1) Conglutin (2S albumin)	17 kDa	Stable to heat treatment and proteolysis, Ara h 2 allergenicity can increase with roasting and decrease with frying or boiling. Hydration before autoclaving increases the effectiveness of the heat treatment by significantly altering its immunoreactivity.	CCNELNEFENNQR	[25,215,219,228,229,232,233,234,235,236]
			NLPQQCGLR	[25,112,216,219,228,229,232,234,236,237]
			CMCEALQQIMENQSDR	[25,215,229,232,235,236,238]
			CQSQLER	[236]
			CDLEVESGGR	[216,229,232,236]
			GAGSSQHQER	[236]
			DEDSYGR	[236]
			ANLRPCEQHLMQK	[236]
			QQEQQFK	[216,236]
			QQWELQGDR	[236]
Ara h 3 (O82580; Q9SQH7) Cupin (Legumin-type, 11S globulin, Glycinin)	60 kDa, 37 kDa (fragment)	The allergenicity of Ara h 3 increases with roasting and decreases with frying or boiling. Hydration before autoclaving increases the effectiveness of the heat treatment by significantly altering its immunoreactivity.	SPDIYNPQAGSLK	[25,42,91,92,143,215,219,229,232,234,235,239]
			RPFYSNAPQEIFIQQGR	[4,20,48,49,92,112,188,237,239]
			FNLAGNHEQEFLR	[25,92,143,144,164,188,219,221,228,229,239]
			LNAQRPDNR	[231,232,237]
			WLGLSAEYGNLYR	[4,25,48,49,50,67,144,164,221,226,229,239]
			SQSENFEYVAFK	[42,50,221,229,233,239]
			AHVQVVDSNGNR	[163,232,239]
			GETESEEEGAIVTVR	[231,239]
			QQPEENACQFQR	[228,234,239]
			TANDLNLLILR	[144,163,164,226,239]
			FFVPPSQQSPR	[239]
			FFVPPFQQSPR	[239]
			NALFVPHYNTNAHSIIYALR	[221,239]
			QIVQNLWGENESEEEGAIVTVR	[239]
			GYFGLIFPGCPSTYEEPAQQGR	[239]
			ADEEEEYDEDEYEYDEEDR	[239]
			VYDEELQEGHVLVVPQNFAVAGK	[221,239]
			TANELNLLILR	[20,112,226,228]
			FFVPPSEQSLR	[226]
			GENESDEQGAIVTVR	[226]
			QIVQNLR	[226]
			AQSENYEYLAFK	[221,226]
			SQSDNFEYVAFK	[221,226]
			TANDNLLLLILR	[221]
			TANELLILILR	[110,221]
			TVNELDLPILNR	[221]
			VYDEELQEGHVLVVPQNFAVAAK	[221]
Ara h 6 (Q647G9) Conglutin (2S albumin)	15 kDa	Stable to heat treatment and proteolysis. Hydration before autoclaving increases the effectiveness of the heat treatment by significantly altering its immunoreactivity.	CDLDVSGGR	[228,234]
Ara h 7 (Q9SQH1; B4XID4; Q647G8) Conglutin (2S albumin)	15 kDa	Stable to heat treatment and proteolysis.	NLPQNCGFR	[228,234]

**Table 3 foods-11-00728-t003:** Overview on the most recent LC-MS methods developed for nuts and peanuts quantification in food products.

Allergen	MS Analyzer (Analysis Mode)	Target Protein	Matrix	LOD	LOQ	Reference
ALMOND	LIT (SRM/SRM^3^)	Pru du 1	Cookies	SRM 17 mg/kg, SRM^3^ 25 mg/kg	SRM 58 mg/kg, SRM^3^ 80 mg/kg	[91]
	QqQ/LIT (SRM)	Pru du 6	Bread	3 mg/kg	/	[49]
	LIT (SRM)	Pru du 1	Biscuits	0.9 mg/kg	3.1 mg/kg	[143]
			Dark chocolate	9 mg/kg	30 mg/kg	
	OrbitrapTM (Full MS)	Pru du 6	Milk chocolate, vanilla ice cream, commercial bread, and breakfast cereals	0.34–1.92 mgPROT/kg, 1.8–10.1 mgNUT/kg	/	[50]
	TQ-S (MRM)	Pru du 6	Incurred chocolate, ice cream, cookies, and sauce	>3 mg/kg	5–25 mg/kg	[52]
	QqQ/LIT (SRM)	Pru du 1	Chocolates	0.4 mg/kg	1.3 mg/kg	[92]
		Pru du 2	Chocolates	0.8 mg/kg	2.6 mg/kg	
HAZELNUT	LIT (SRM/SRM^3^)	Cor a 9	Cookies	SRM 30 mg/kg, SRM^3^ 35 mg/kg	SRM 90 mg/kg, SRM^3^ 110 mg/kg	[91]
	QqQ/LIT (SRM)	Cor a 9	Bread	5 mg/kg	/	[49]
	LIT (SRM)	Cor a 9	Biscuits	1.3 mg/kg	4.5 mg/kg	[143]
			Dark chocolate	14 mg/kg	49 mg/kg	
	LIT (SRM)	Cor a 9	Cookies	1–5 mg/kg	16–32 mg/kg	[218]
	TQ-S (MRM)	Cor a 9	Incurred chocolate, ice cream, cookies, and sauce	>3 mg/kg	2.5–5 mg/kg	[52]
	Q-Orbitrap (t-SIM/dd2)	Cor a 9	Incurred cookie	4 mg/kg	12 mg/kg	[114]
	QTRAP 6500 (IDA-MS/MS)	Cor a 9	Cookies	≤2.25 mg/kg	≤3 mg/kg	[4]
			Ice cream	≤2 mg/kg	≤10 mg/kg	
			Breakfast cereal	≤0.45 mg/kg	≤3 mg/kg	
			Milk chocolate	≤10 mg/kg	≤30 mg/kg	
	QqQ/LIT (SRM)	Cor a 9	Chocolates	0.5 mg/kg	1.7 mg/kg	[92]
	TripleTOF 6600 (IDA-MS/MS)	Cor a 9	Cookies	3.1 mg/kg	5.3 mg/kg	[67]
CASHEW	LIT (SRM/SRM^3^)	Ana o 2	Cookies	SRM 14 mg/kg, SRM^3^ 30 mg/kg	SRM 46 mg/kg, SRM^3^ 98 mg/kg	[91]
	LIT (SRM)	Ana o 2	Biscuits	0.5 mg/kg	1.6 mg/kg	[143]
			Dark chocolate	15 mg/kg	50 mg/kg	
	Orbitrap^TM^ (Full MS)	Ana o 2	Milk chocolate, vanilla ice cream, commercial bread, and breakfast cereals	0.78–2.02 mgPROT/kg 4.3–11.2 mgNUT/kg	/	[50]
	QqQ/LIT (SRM)	Ana o 2	Chocolates	0.7 mg/kg	2.3 mg/kg	[92]
	TQ-S (MRM)	Ana o 2	Incurred chocolate, ice cream, cookies, and sauce	>3 mg/kg	2.5 mg/kg	[52]
		Ana o 3		>3 mg/kg	2.5 mg/kg	
PISTACHIO	Orbitrap^TM^ (Full MS)	Pis v 2	Milk chocolate, vanilla ice cream, commercial bread, and breakfast cereals	1.28–1.90 mgPROT/kg 7.10–10.6 mgNUT/kg	/	[50]
		Pis v 5		0.91–1.38 mgPROT/kg 5.1–7.6 mgNUT/kg	/	
	TQ-S (MRM)	Pis v 2, Pis v 5	Incurred chocolate, ice cream, cookies, and sauce	>3 mg/kg	2.5 mg/kg	[52]
	QqQ/LIT (SRM)	Pis v 2	Chocolates	0.4 mg/kg	1.3 mg/kg	[92]
WALNUT	LIT (SRM/SRM^3^)	Jug r 4	Cookies	SRM 55 mg/kg, SRM^3^ 50 mg/kg	SRM 180 mg/kg, SRM^3^ 160 mg/kg	[91]
	QqQ/LIT (SRM)	Jug r 1	Bread	70 mg/kg	/	[49]
	LIT (SRM)	Jug r 4	Biscuits	0.8 mg/kg	2.6 mg/kg	[143]
			Dark chocolate	5 mg/kg	18 mg/kg	
	Orbitrap^TM^ (Full MS)	ND	Milk chocolate, vanilla ice cream, commercial bread, and breakfast cereals	0.80–5 mgPROT/kg 5.7–35.7 mgNUT/kg	/	[50]
	TQ-S (MRM)	Vici lin-like protein	Incurred chocolate, ice cream, cookies, and sauce	>3 mg/kg	12.5 mg/kg	[52]
		Jug r 1			5 mg/kg	
	QqQ/LIT (SRM)	Jug r 2	Chocolates	0.6 mg/kg	2.0 mg/kg	[92]
PEANUT	LIT (SRM/SRM^3^)	Ara h 3/4	Cookies	SRM 10 mg/kg, SRM^3^ 27 mg/kg	SRM 37 mg/kg, SRM^3^ 90 mg/kg	[91]
	QqQ/LIT (SRM)	Ara h 1	Bread	11 mg/kg	/	[49]
	LIT (SRM)	Ara h 3/4	Biscuits	0.1 mg/kg	0.3 mg/kg	[143]
			Dark chocolate	7 mg/kg	25 mg/kg	
	LIT (SRM)	Ara h 1	Cookies	8–9 mg/kg	30 mg/kg	[218]
	TQ-S (MRM)	Ara h 2, Ara h 3/4	Incurred chocolate, ice cream, cookies, and sauce	>3 mg/kg	5 mg/kg	[52]
	Q-Orbitrap (t-SIM/dd2)	Ara h 1	Incurred cookie	7 mgPROT/kg	24 mgPROT/kg	[114]
	QTRAP 6500 (IDA-MS/MS)	Ara h 3	Cookies	≤0.3 mg/kg	≤3 mg/kg	[4]
			Ice cream	≤2 mg/kg	≤10 mg/kg	
			Breakfast cereal	≤0.1 mg/kg	≤1 mg/kg	
			Milk chocolate	≤1.5 mg/kg	≤10 mg/kg	
	QqQ/LIT (SRM)	Ara h 1	Chocolates	0.8 mg/kg	2.6 mg/kg	[92]
		Ara h 3/4	Chocolates	1.3 mg/kg	4.3 mg/kg	
	Q-TOF (MS/MS)	Ara h 1	Cookies	<2.5 mg/kg	0.30 mg/kg	[216]
		Ara h 2			0.13 mg/kg	
	TripleTOF 6600 (IDA-MS/MS)	Ara h 3	Cookies	2.2 mg/kg	6.7 mg/kg	[67]
		Ara h 3	Breakfast cereal	1.2 mg/kg	2.0 mg/kg	
	QqQ (MRM)	Ara h 1	Wheat flour matrix (both raw and cooked)	0.15 mg/kg	0.31 mg/kg	[217]

## 3. Conclusions

In the present review, an overview of the four most widespread potentially allergenic nuts, including hazelnuts, walnuts, cashews, and pistachios, and of almonds and peanuts has been provided. These products represent a relevant threat for the allergic population, due to cross-contamination that is likely to occur during manufacturing in the food plants. Both the biochemical characterization of the allergenic proteins for each ingredient and the methodological approaches so far developed for their detection, based on immunochemistry or mass spectrometry, have been exhaustively described. Finally, with regard to the MS methods, a list of the most reliable and sensitive markers has been also delivered for each food allergen, starting from peer-reviewed scientific literature, proposing markers for tracing and quantifying the six allergens in complex and processed food commodities. Efforts of the scientific community, along with the most recently funded projects on this regard, have been also presented in the perspective of reaching a consensus on a common and harmonized analytical approach for multiple allergen quantification in complex food matrices.

## Figures and Tables

**Figure 1 foods-11-00728-f001:**
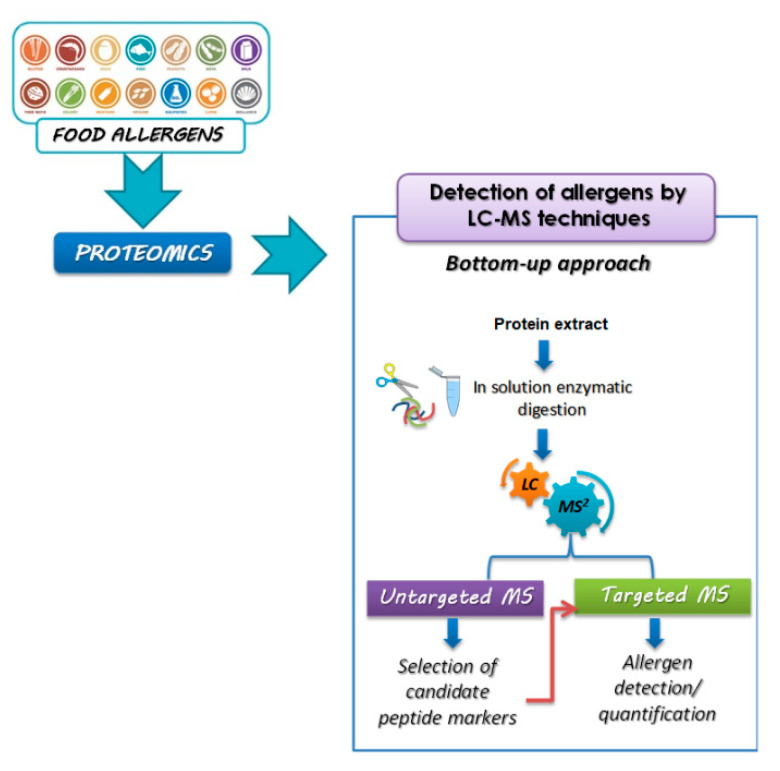
Overview of proteomic pipelines based on *bottom-up* strategies for food allergen quantification.

**Figure 2 foods-11-00728-f002:**
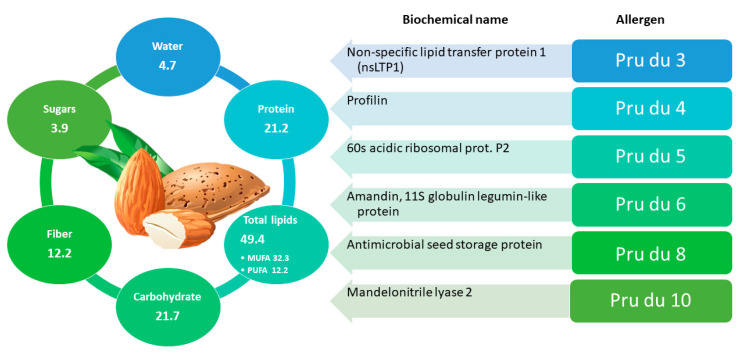
(**Left**) Relevant information on the nutritional composition (g/100 g) relating to the *Prunus dulcis* species. (**Right**) list of allergenic proteins identified and registered to date for *Prunus dulcis* on the WHO/IUIS online allergen nomenclature platform.

**Figure 3 foods-11-00728-f003:**
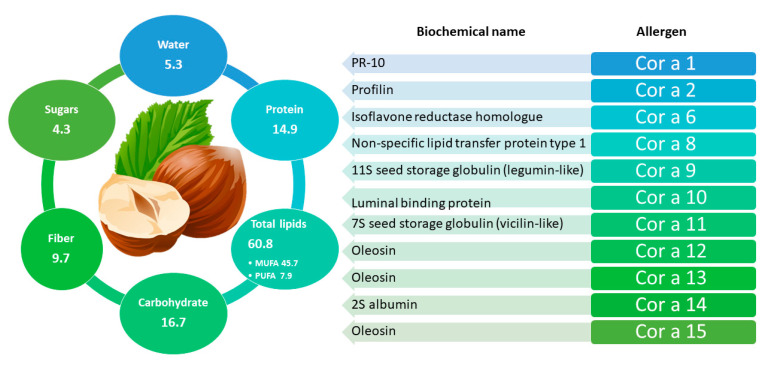
(**Left**) relevant information on the nutritional composition (g/100 g) relating to the *Corylus avellana* species. (**Right**) list of allergenic proteins identified and registered to date on the WHO/IUIS online allergen nomenclature platform.

**Figure 4 foods-11-00728-f004:**
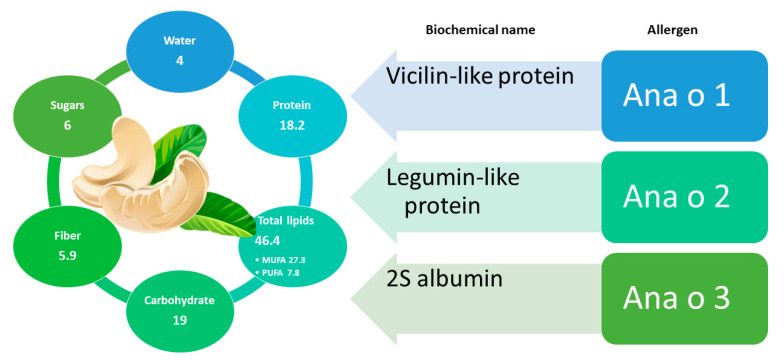
(**Left**) relevant information on the nutritional composition (g/100 g) relating to the *Anacardium occidentale* species. *(***Right**) list of allergenic proteins identified and registered to date on the WHO/IUIS online allergen platform.

**Figure 5 foods-11-00728-f005:**
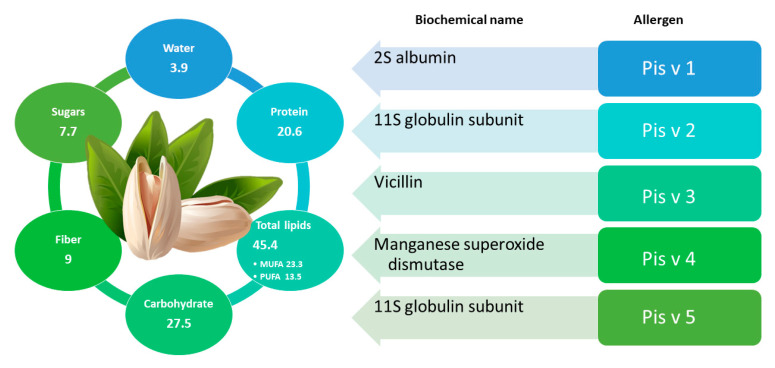
(**Left**) relevant information on the nutritional composition (g/100 g) relating to the *Pistacia vera* species. (**Right**) list of allergenic proteins identified and registered to date on the WHO/IUIS online allergen platform.

**Figure 6 foods-11-00728-f006:**
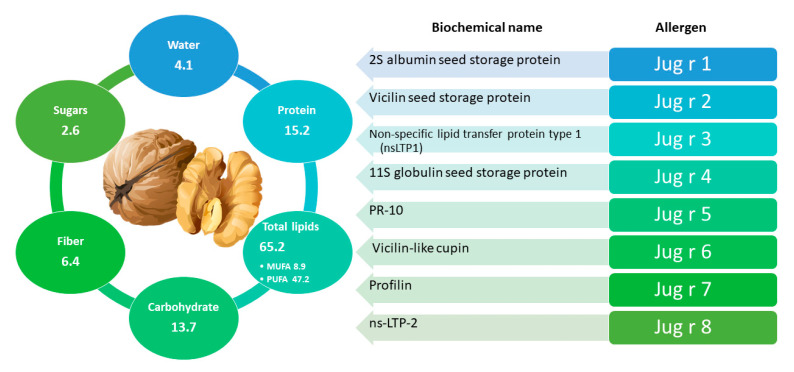
On the left there is the relevant information on the nutritional composition (g/100 g) relating to the *Juglans regia* species. On the right is the list of allergenic proteins identified and registered to date on the WHO/IUIS online allergen platform.

**Figure 7 foods-11-00728-f007:**
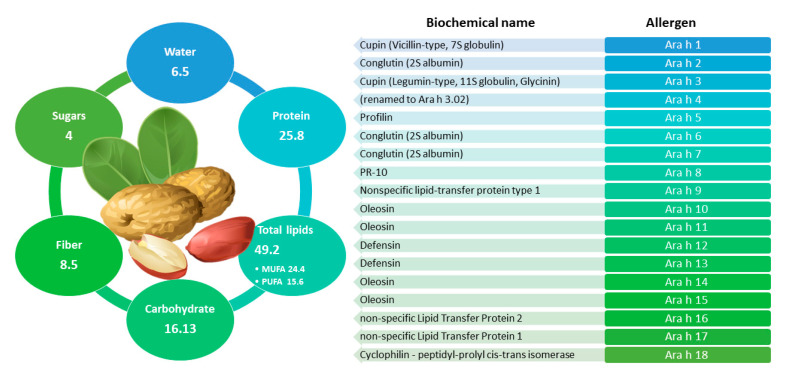
(**Left**) relevant information on the nutritional composition (g/100 g) relating to the *Arachis hypogaea* L. species. (**Right**) list of allergenic proteins identified and registered to date on the WHO/IUIS online allergen platform.

## Data Availability

Not applicable.

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
