# Peer review of "Tree Nuts and Peanuts as a Source of Beneficial Compounds and a Threat for Allergic Consumers: Overview on Methods for Their Detection in Complex Food Products"

_foods, 2022, doi:10.3390/foods11050728_

Round 1

Reviewer 1 Report

Foods-16066625

Review

Tree nuts and peanuts as a source of beneficial compounds and a threat for allergic consumers: overview on methods for their detection in complex food products

The manuscript aims at covering both the biochemical aspect linked to the identified allergenic proteins for each allergen category and the different methodological approaches developed for allergens detection and identification. Attention has been also paid to MS methods and to current efforts of the scientific community to identify a harmonized approach for allergens quantification through detection of allergen markers.

In the recent past, several studies have focused on genetic, epigenetic and environmental risk factors related to food allergies, bringing more clarity on these issues and opening interesting perspectives in terms of improvement of prevention and treatment strategies targeted to individuals at risk. This is one f its kind review addressing the needs.

The review was well written

Figures and tables were adequate, with good discussion and necessarily concluded.

Tables were informative,

Correct “quantification trough detection of allergen markers” ……..

The literature review was also adequate.

English/grammar has to be checked once again.

Should minor mandatory revision.

Author Response

see the atteched file please

Reviewer 2 Report

I have read the review 'Tree nuts and peanuts as a source of beneficial compounds and a threat for allergic consumers: overview on methods for their detection in complex food products' with much interest. The detection of major allergens in food stuff is extremely important for the well-being of peanuts and nut allergic patients. This is closely related to using the correct and most sensitive detection method. This review gives a thorough overview of the methods currently used for different food matrices (complex or processed).

Minor comment: 

Abstract line 23: 'MS' better not use abbreviations in the abstract

Introduction 1.1 Line 42-43: 'Moreover, approximately 20-30% of people suffering.......' the used reference mainly discusses the mono- or multi-sensitization to peanut/tree nuts and not specifically cross-reactivity, please check if correct ref is used

Introduction 1.2 Line 68-69: The sentence 'in order to protect.....' does not read well, please check grammar. Line 74: change 'reviewingn' to reviewing'. Line 75-77: 'At European level......' does not read well, please check grammar.

Introduction 1.3 Line 130-132: 'Despite its advantages....' this sentence does not seem to be complete or the , should be changed to .

Introduction 1.4 Line 229: 'sensitivity achieved was not very challenging'. What is meant by 'challenging'? The word challenging is used more often in the manuscript without a clear purpose or explanation, please adjust. Line 281: 'total food allergen protein, ' is missing after protein. Line 302 (a method) and line 306 (a reference method), please elaborate on what kind of method.

2.1.2 Line 397: When describing a specific ELISA please ad the type of ELISA (direct, indirect, competitive etc). Sandwich ELISA is always defined, however throughout the manuscript other ELISA's are not always defined, please correct.

2.1.3 Line 425/433/435 Almond (Pru 1) should be Pru du 1). Line 432-433: LOD and LOQ abbreviations have been explained previously, only use abbreviation here.

3.1.2 Line 534-536: In 2021, to improve manufacturing.....' is a very long sentence and not grammatically correct. Please rewrite.

4.1.2 Line 670: 'Development of ELISAs specific to Ana o 3 very low cross-reactivity.......' Add 'show' between Ana o 3 and very.

5.1.1 Line 751: Change 'Ana or 3' to 'Ana o 3'

6.1.1 Line 859: heat treatments above 90C for long times. What is 'long times'? Please specify

6.1.2 Line 881: only use abbreviation ELISA, explained previously

7.1.1 Line 981 Use abbreviation for Non-specific lipid transfer proteins, explained previously

7.1.2 Line 1009: delete 'enzyme immunoassay'

7.1.3 Line 1078: Use abbreviation MS, explained previously. Line 1087: 'chocolates spiked with allergenic'. Spiked with allergenic what?

Author Response

pleasse see answers in the file attached
